# A Comparative Bioinformatic Investigation of the Rubisco Small Subunit Gene Family in True Grasses Reveals Novel Targets for Enhanced Photosynthetic Efficiency

**DOI:** 10.3390/ijms26157424

**Published:** 2025-08-01

**Authors:** Brittany Clare Thornbury, Tianhua He, Yong Jia, Chengdao Li

**Affiliations:** 1Western Crop Genetics Alliance, College of Science, Health, Engineering and Education, Murdoch University, 90 South Street, Murdoch, WA 6150, Australia; bcrobertson97@gmail.com (B.C.T.); tianhua.he@murdoch.edu.au (T.H.); y.jia@murdoch.edu.au (Y.J.); 2Western Australian State Agricultural Biotechnology Centre, Murdoch University, 90 South Street, Murdoch, WA 6150, Australia; 3Department of Primary Industries and Regional Development, Western Australian Government, 3-Baron-Hay Court, South Perth, WA 6151, Australia

**Keywords:** photosynthesis, RuBisCO, barley, C3 crops, C4 crops, cereals, grasses, crop improvement

## Abstract

Ribulose bisphosphate carboxylase (RuBisCO) is the primary regulator of carbon fixation in the plant kingdom. Although the large subunit (RBCL) is the site of catalysis, RuBisCO efficiency is also influenced by the sequence divergence of the small subunit (RBCS). This project compared the RBCS gene family in C3 and C4 grasses to identify genetic targets for improved crop photosynthesis. *Triticeae*/*Aveneae* phylogeny groups exhibited a syntenic tandem duplication array averaging 326.1 Kbp on ancestral chromosomes 2 and 3, with additional copies on other chromosomes. Promoter analysis revealed a paired I-box element promoter arrangement in chromosome 5 RBCS of *H. vulgare*, *S. cereale*, and *A. tauschii*. The I-box pair was associated with significantly enhanced expression, suggesting functional adaptation of specific RBCS gene copies in *Triticaeae*. *H. vulgare*-derived pan-transcriptome data showed that RBCS expression was 50.32% and 28.44% higher in winter-type accessions compared to spring types for coleoptile (*p* < 0.05) and shoot, respectively (*p* < 0.01). Molecular dynamics simulations of a mutant *H. vulgare* Rubisco carrying a C4-like amino acid substitution (G59C) in RBCS significantly enhanced the stability of the Rubisco complex. Given the known structural efficiency of C4 Rubisco complexes, G59C could serve as an engineering target for enhanced RBCS in economically crucial crop species which, in comparison, possess less efficient Rubisco complexes.

## 1. Introduction

As the sole carbon-fixing catalyst in the Tree of Life, ribulose bisphosphate carboxylase (RuBisCO) is responsible for converting most of the atmosphere’s inorganic carbon (over 90%) into biomass [1]. Consequently, RuBisCO is the only food producer sustaining the global food chain and is arguably one of the most essential enzymes on Earth. RuBisCO acts as a processing enzyme in the initial step of the light-independent reactions of photosynthesis, encompassing the Calvin cycle [2]. Its catalytic action facilitates the incorporation of one carbon from atmospheric CO_2_ into a five-carbon substrate, ribulose-1,5-bisphosphate (RuBP), resulting in the production of two molecules of 3-phosphoglyceric acid (3PGA), each being a three-carbon product [3]. These molecules then participate in reactions that lead to the eventual generation of more complex sugar molecules (i.e., glucose), which can be used as an energy source [4].

RuBisCO is the entry point for inorganic carbon in subsequent processing reactions and constitutes the rate-limiting step of carbon fixation. It is thus the central determinant of an organism’s carbon-fixation efficacy [5]. Activation of the RuBisCO complex for catalysis involves both carbamylation and stabilisation of the active site via magnesium ion binding [6]. Without these processes, enzymatic activity is blocked—a likely safeguard against energy wastage resulting from RuBisCO inefficiency [7]. The inefficiency of RuBisCO is unusual, given its ubiquity and necessity as the prime inorganic carbon assimilator. Its double affinity for both CO_2_ and O_2_ poses substantial challenges to the enzyme’s carbon fixation efficiency, which is exacerbated by increases in temperature [5]. As temperatures rise, the reduced solubility of CO_2_ relative to O_2_ results in increased oxygen concentration in the leaf cellular space. This, combined with a higher specificity toward oxygen at elevated temperatures, leads to the preferential binding of oxygen by RuBisCO and subsequent ATP wastage in photorespiration [8,9].

RuBisCO operates as a moderately complex molecular machine, with the predominant Form I RuBisCO making up a substantial enzymatic complex of approximately 550 kilodaltons (kDa). This complex comprises eight small (RBCS) and eight large subunits (RBCL), forming a hexadecameric structure [10]. The catalytic activity occurs in the large subunits, which pair as dimers and create an α/β barrel-shaped arrangement at the C-terminal domain [11,12]. However, the observed diversity of RBCL is minimal across plant genera, exhibiting a high degree of conservation even among distantly related groups. This is demonstrated by species with group 1B RuBisCOs (from higher plants) showing an average sequence identity of 90% or more in the large subunit [13,14]. In contrast, sequence identity of as little as 30% is observed among plant RBCS, with copy numbers ranging from 1 to over 20 in some species [14,15].

Carbon fixation capacity variability is prominent among plant RuBisCOs [16,17,18,19]. This has been partly attributed to sequence divergence of the small subunit. The impact of specific RBCS amino acid substitutions on RuBisCO catalysis has provided promising evidence that RBCS mutagenesis drives RuBisCO evolution. The result is an improved capacity for RuBisCO to function in ‘rate-limiting’ environments across select plant taxa. One such study recently assessed the impact of hybridisation between RBCL derived from rice and RBCS derived from *Sorghum bicolor*, a C4 crop known for its high photosynthetic efficacy at elevated temperatures. The observed effect of hybridisation resulted in a higher catalytic rate (*K*_cat_), attributed to residue L102 in the βC-βD Hairpin of sorghum-derived RBCS, leading to modified flexibility of the 60S loop in RBCL [20]. These results support the notion that RBCS sequence identity can fine-tune the catalytic turnover of RuBisCO, providing accelerated pathways to achieve bioengineering targets for enhanced biomass generation.

Despite C3 plants dominating a significant portion of the global market, their RuBisCOs are considerably more susceptible to the harmful effects of photorespiration. This is both due to the reduced catalytic efficiency of the RuBisCO complex and the absence of C4-adapted cellular compartmentalization to mitigate O_2_ specificity issues [17,21,22,23,24]. Given the known catalytic advantages of C4 RuBisCOs, this project aimed to investigate the evolutionary and structural characteristics of true grass RuBisCO small subunit (*RBCS*) gene families (*n* = 16) to identify novel targets for photosynthetic improvement. The successful engineering of RuBisCO complexes in agriculturally significant crop species could offer a promising avenue to boost productivity targets in a future likely to be characterised by extensive global warming events.

## 2. Results

### 2.1. The RBCS Gene Family Exhibits a Distinct Copy Number Separation Between Phylogenetic Groupings Strongly Correlated with Genome Size

Phylogenetic comparison of *RBCS* gene sequences in the study species indicated the presence of a multigene family across all representatives except *S. bicolor*, which possessed a single *RBCS* gene copy (Figure 1A and Figure 2A). The most basal grass lineage in our study (i.e., *P. latifolius*) similarly possessed a low *RBCS* copy number (*n* = 2) along with the basal *Bambusoideae* subfamily (with 4 and 3 *RBCS* gene copies identified for *O. latifolia* and *R. guianensis*, respectively). This potentially suggests that an ancestral low *RBCS* copy number state existed in the grasses, that generally expanded as species radiation occurred over time (Figure 1A and Figure 2A). A somewhat noticeable divergence in *RBCS* copy numbers was identified between C3 and C4 species, with average *RBCS* copy numbers of 6.6 and 3.17, respectively (Figure 2C). When classifying *RBCS* copy number by climate type, warm climate-adapted species exhibited an average *RBCS* copy number of 3.3. In contrast, cold climate-adapted species in this study, of which the majority of species (83.4%) consisted of *Triticeae*/*Aveneae* tribe members, possessed an average copy number of 8.67 (Figure 2B). A large expansion of the *RBCS* gene family was apparent for members of the *Triticeae* and *Aveneae* tribes, with a minimum copy number ranging from 7 (observed for *A. tauschii*) to 11 copies (observed in *S. cereale* and *A. atlantica*). All other grass representatives in our study possessed an *RBCS* copy number less than or equal to 5 (Figure 1A). *RBCS* gene copies in *A. atlantica* (group IV; Figure 1A) formed a separate cluster from those in the *Triticeae* tribe. This observation indicated two separate *RBCS* gene family expansion events occurred, and these likely happened following the divergence of the *Triticeae* and *Aveneae*. Furthermore, three distinct *RBCS* gene family clusters were present within the *Triticeae* phylogenetic group. Unique clusters for *RBCS* on chromosome 2 were observed for all *Triticeae* members (group I; Figure 1A), *RBCS* on chromosome 1 for *S. cereale* and *H. vulgare* (group II; Figure 1A) and *RBCS* on chr5 for all *Triticaeae*, these being *S. cereale*, *H. vulgare*, *T. urartu*, and *A. tauschii* (Group III; Figure 1A). These observations, in turn, suggested that the expanded *RBCS* gene family was inherited via a common ancestor before diversification of *Triticeae* species members. Additional *RBCS* copies on chr7 were observed for *S. cereale* in the group I cluster (Figure 1A).

To further assess the source of *RBCS* gene family expansion in members of the *Triticeae*/*Aveneae* tribes, Pearson correlation analysis was carried out to determine the extent of correlation between genome size (Mbp) and *RBCS* copy number. Overall, genome size was highly positively correlated with *RBCS* copy number in the species compared (R = 0.81, *p* < 0.001), suggesting that *RBCS* expansion in the *Triticaeae/Aveneae* could be attributed to genome duplication events at some point in evolutionary history (Figure 1B). However, *Z. mays*, which possesses a relatively large genome compared to other members of the C4 group, exhibits a somewhat conflicting relationship between genome size and *RBCS* copy number. In this instance, despite *Z. mays* possessing a 2400 Mbp genome (which is only ~35% smaller than the genome size of *A. atlantica* (3720 Mbp)), the *Z. mays RBCS* copy number (*n* = 2) is ~5.5 times smaller than that of the *RBCS* gene family possessed by *A. atlantica*. This may suggest that *RBCS* copy number does not substantially affect photosynthetic processes in some C4 grass lineages, perhaps due to possessing enzymes with high catalytic efficacy. This observation may also substantiate the low *RBCS* copy number prevalent in *S. bicolor*. Despite possessing a somewhat equivalent genome size (732 Mbp) to other species in this study, such as *R. guianensus* (626 Mbp, RBCS copy number = 3) and *O. latifolia* (646 Mbp, *RBCS* copy number = 4), *S. bicolor* possessed a drastically reduced *RBCS* gene family size, consisting of a single gene copy.

Expanding into the *H. vulgare* (barley) pangenome, *RBCS* copy number was mostly uniform across accessions, with 83% of varieties exhibiting an *RBCS* copy number of 10 (Figure 2D,E). When comparing accessions based on domestication status, only one cultivar (Barke) possessed a divergent *RBCS* copy number (*n* = 9) (Figure 2D). No landraces possessed an *RBCS* copy number lower than the dominant 10 RBCS copies. However, seven accessions exhibited one extra *RBCS* copy—these being 10TJ18, HOR10892, HOR13594, HOR1702, HOR18321, HOR19184 and HOR8148 (Figure 2D). In addition, one landrace accession (HOR7552) showed an *RBCS* copy number of 12—the highest observed in the pangenome varieties studied. Wild barley accessions also showed some diversity in *RBCS* copy number, with the lowest copy number recorded for FT333 (*n* = 8), followed by HID249 (*n* = 9), and WBDC103 exhibiting the greatest copy number (*n* = 11) (Figure 2D). When classified based on geographic origin, the greatest *RBCS* copy number diversity was observed for the Middle East accessions (ranging from 8 to 11 copies). Finally, geographic regions with accessions exhibiting predominantly high *RBCS* copy numbers included south Asia (with 40% of varieties possessing *RBCS* copy number > 10) and Central Asia, of which both representatives possessed an *RBCS* copy number of 11 (Figure 2E).

### 2.2. The Expanded RBCS Gene Family Shows Syntenic Associations Between Representatives of the Triticeae/Aveneae

Given the phylogenetic relationships observed between the *Triticae*/*Aveneae RBCS* gene families (covered in Section 2.1), synteny analysis was carried out to establish a clearer understanding of *RBCS* gene family expansion in *H. vulgare* (barley), *T. urartu* (diploid wheat), *S. cereale* (rye), *A. tauschii* (Aegilops/Tausch’s Goatgrass), and *A. atlantica* (diploid oat). Initial assessment of *RBCS* gene loci revealed that for those gene copies located on chromosome 2 in the *Triticeae* (and the syntenic chromosome 3 in *A. atlantica*), the expanded *RBCS* gene family was present as a compact tandem duplication array. The array spanned an average of 326.1 Kbp, and presented as small as 52.4 Kbp in *A. tauschii*. The proximity of *RBCS* gene copies in turn suggested that these arrays were likely expressed together in the respective species. Following the assessment of macrosynteny blocks, we determined the most likely chromosomal origin of the large *RBCS* gene family in *Triticeae*/*Aveneae* members. Synteny analysis revealed that *RBCS* copies on Chr1 were syntenic between *H. vulgare* and *S. cereale*, and *RBCS* copies on Chr5 were syntenic between *H. vulgare*, *T. urartu*, *S. cereale*, and *A. tauschii* (Figure 3A–C). The syntenic relationships between *RBCS* gene copies in *Triticeae* species on Chr1 and Chr5, echoed the pattern of phylogenetic clustering observed between group II and III *RBCS* gene groupings (covered in Section 2.1).

*RBCS* gene copies on chromosome 2 of *H. vulgare* exhibited syntenic relationships between all members of the *Triticeae*, in addition to *A. atlantica* of the *Aveneae* tribe. As shown by Figure 3D, all species demonstrated a syntenic link between at least two *RBCS* gene copies on chr2/3 with *RBCS* copies present in the tandem duplication array on chromosome 2 in *H. vulgare*. The combined results of synteny analysis suggested that diversification of the *RBCS* gene family in the *Triticeae*/*Aveneae*, was achieved via the expansion of *RBCS* gene copies from an ancestral arrangement of two *RBCS* copies on chromosome 2/3, in a common progenitor grass species to both lineages.

### 2.3. Cis-Regulatory Element Profiles of Grass Family RBCS Are Dominated by a Light- and Drought-Receptive Landscape

A cross-comparison of cis-regulatory element (CRE) configurations among the true grasses showed highly analogous CRE composition. *RBCS* promotors were primarily dominated by drought-inducible elements in most species studied, representing on average approximately 26.4% of CRE function (Figure 4). *RBCS* promoters additionally exhibited a high prevalence of light-inducible CREs, constituting approximately 20.6% of CRE function. Light-inducible elements were the dominant CRE detected in *RBCS* promoters of the C4 members *Z. mays*, *U. fusca*, and *O. thomaeum*, potentially suggesting evolutionary trends towards enhanced light-dependent regulation in these species (Figure 4B). Other prominent CRE identified in the *RBCS* promoter landscape include those related to ABA-response, Methyl Jasmonate (MeJA) response, and multiple stress response pathways, representing on average 7.7%, 6.7% and 8.34% of CRE function, respectively, across species possessing the CRE type(s). A more defined comparison of relative CRE composition across species showed some unique trends. Circadian CREs predominated the *RBCS* promoters of C3 grasses, being present in at least one *RBCS* gene promoter in all members of the *Triticeae* (with the exception of *S. cereale*) (Figure 4A). CREs with strong prevalence across all studied *RBCS* grass promoters included MYC (absent only from *AvatRBCS6* and *RgRBCS1*), MYB (absent only from *BdRBCS4* and *RgRBCS1*), and STRE, further highlighting that *RBCS* gene families are regulated by a variety of stress response pathways. The light-associated G-box and I-box elements were also prevalent in the majority of *RBCS* promoters, present in 90% and 73.4% of upstream gene regions, respectively. The low-temperature response element (LTR) was identified in the *RBCS* promoters of all species except *P. latifolius* and was present in 63.4% of all *RBCS* gene promoters studied, showing no clear differentiation between the cool and warm climate grasses. LTR element distribution likely highlights an intrinsic interplay between light intensity and, thus, temperature sensitivity in regulating *RBCS* expression. ARE elements involved in gene expression under low oxygen conditions were present at high levels across all species studied, with a prevalence of 87.8%, indicating an intrinsic relationship between light-sensitive *RBCS* genes and control of anaerobic respiration. Finally, CREs involved in wound healing had a notable prevalence across *RBCS* promoters such as WUN, with the WRE3 elements being present in the *RBCS* gene family of all grass species in this study. This in turn conveyed a relationship between regulation of carbon assimilation and wound healing. CRE function and distribution in the *RBCS* gene family of the true grasses paint a landscape defined by drought and light receptivity, with links to stress-associated pathways and those bound to wound healing and anoxic gene expression. This is a pattern not too unconventional, considering the role of *RBCS* in biomass accumulation during photosynthesis.

### 2.4. Unique Distribution of Terminal Cis-Regulatory Elements in the RBCS Promoters of Select Members of the Triticeae Exhibit Concordantly Enhanced Expression

This study investigated potential associations between cis-regulatory element configuration and *RBCS* gene family expression. Isolating six major CREs frequently reported in *RBCS* promoters, we sought to identify expression profiles that may be influenced by CRE arrangements. Following CRE mapping of the 2000 bp upstream region of *RBCS* gene(s), we identified a region close to the start of the gene in *RBCS* gene copies located on chromosome 5 in *H. vulgare* (*HvRBCS1* and *HvRBCS5*), *S. cereale* (*ScRBCS1* and *ScRBCS6*), and *A. tauschii* (*AtauRBCS1* and *AtauRBCS6*), containing an I-box element pair with elements spaced ~25 bp apart (Figure 5; Appendix A)). In all promoter regions containing the pair, it was located no further than 168 bp from the gene start site. As shown by the phylogenetic tree in Figure 5, all gene copies possessing the terminal I-box pair were closely related, reflecting the likely inheritance of the pair from a common ancestor shared by the three *Triticeae* members. Following transcriptomic analysis of *RBCS* gene expression in *H. vulgare*, *S. cereale* and *A. tauschii*, the chromosome 5 *RBCS* showed clearly differentiated expression, with a transcript abundance representing at least over 20% of total gene expression per chromosome five gene copy, and up to 48.7% of total *RBCS* transcript abundance in *A. tauschii* for a single gene copy (*AtauRBCS1*) (Figure 5). In comparison, all other *RBCS* gene family copies per species, each represented under 10% of total *RBCS* transcript abundance when considered individually, with *ScRBCS2* representing as little as 0.05% of the total *RBCS* transcript abundance in *S. cereale* (Figure 5). Our observations indicated that a likely ancestrally derived expression profile has been retained in *H. vulgare*, *S. cereale* and *A. tauschii*, with *RBCS* copies on chromosome 5 dominating most of the transcript output. The remaining chromosome copies, despite altogether representing about half of the total *RBCS* transcript abundance in the three species, each exhibit, on average, about 4.36% of the total *RBCS* transcript abundance. These trends suggest a divergent function of *RBCS* expression in a chromosome-specific pattern, which may allow enhanced fine-tuning of *RBCS* expression via the regulation of gene subsets for select members of the *Triticeae*.

### 2.5. Divergent RBCS Expression Profiles Are Observed Between H. vulgare Pangenome Accessions Possessing a Spring and Winter-Type Growth Habit

Utilising data provided by the barley pangenome consortium, we investigated *RBCS* transcript expression for 19 barley varieties to determine any trends between transcript abundance and growth habit over five tissue types (Caryopsis, Coleoptile, Inflorescence, Root and Shoot). As expected for a photosynthetic gene, expression levels were highest in the coleoptile and shoot, with average ‘transcripts per million’ (TPM) of 7102 and 9539, respectively, for winter-type varieties (*n* = 5) and 3529 and 6826 for spring-type varieties (*n* = 14) (Figure 6). Both expression profiles for the coleoptile and shoot showed a statistically significant difference in expression, with winter-type varieties exhibiting 28.44% higher transcript levels in the shoot (*p* < 0.01) and 50.32% higher transcript levels in the coleoptile versus their spring-type counterparts (*p* < 0.05) (Figure 6). Winter type expression also exceeded spring type expression in the caryopsis (with 24.8% higher expression) (*p* < 0.05) and inflorescence, with 17% higher expression—however, the difference in transcript abundance observed for the inflorescence was not statistically significant. The difference in root expression between the spring and winter type varieties was minimal (10%), with highly reduced expression for both growth habits (TPM value < 4) (Figure 6). The above expression data is suggestive that adaptive regulation of the *RBCS* gene family is present in barley varieties with a winter-type growth habit, perhaps as a compensatory mechanism due to exposure to low sunlight environments.

### 2.6. Barley Mutant Rubisco Possessing Amino Acid Substitution Derived from Wildtype Maize RBCS Exhibits Superior Protein Stability

Molecular dynamics analysis was completed in this study to better understand the perceived differences in RuBisCO efficiency observed between different grass species. Given that the small subunit exhibits substantially higher diversity than RBCL, we investigated the effect of two amino acid substitutions on the RBCS sequence of *H. vulgare*, L83Y and G59C, noting that both represent the native amino acid at each position in the C4 crop *Z. mays* (Figure 7A,B). The selection of the candidate amino acid substitutions was a multi-step process. First, candidate residues in the small subunit were identified within 4 angstrom units of the large subunit. An amino acid alignment of the RBCS sequences from the 16 true grass species was then compared to determine the nature of amino acid sequence divergence at the identified sites. Following alignment, the L83Y substitution was identified as a desirable candidate. This was because 100% of C4 species in this study possessed the aromatic Y residue at this position versus only one of the C3 representatives (*O. latifolia*), with all other C3 RBCS sequences instead primarily possessing the aliphatic L or I residue. The C residue at position 59 showed prevalence only among C4 representatives, with *Z. mays*, *S. bicolor*, and *S. italica* possessing the G59C substitution (versus all other grass representatives possessing G at position 59). The protein alignment thus showed that cysteine (C) at position 59 was absent in all C3 photosynthesizing grasses included in this study. In addition, G59C was an amino acid substitution of interest identified following proximity-based selection of candidate RBCS residues close to the large subunit.

Selected residues were then assessed with the Sorting Intolerant From Tolerant (SIFT) sequence-based predictor. Given that SIFT scores ranging from 0.0 to 0.05 are predicted to be deleterious to protein function, and those >0.05 up to 1.0 are tolerated, the G59C substitution was predicted to be tolerated, with a score of 0.07, and the L83Y substitution was predicted impact protein function with a score of 0.04. It should be noted that due to the low nature of the SIFT score for the G59C substitution (0.07—near the 0.05 threshold), we additionally chose to assess impacts on protein stability for this substitution. DynaMut2 results indicated that both substitutions somewhat impact stability, with the L83Y (ΔΔG Stability: −1.45 kcal/mol) substitution predicted to have a more significant impact on protein stability than the G59C substitution (ΔΔG Stability: −0.56 kcal/mol) (in conjunction with the SIFT score results).

Next, we completed molecular dynamics analysis of the wildtype RBCS (complexed with RBCL) derived from *H. vulgare*. We also modelled *H. vulgare* RBCL complexed with a C3-derived RBCS from *O. sativa* and a C4-derived RBCS from *Z. mays.* Finally, two additional in silico generated RBCS mutants were modelled using the *H. vulgare* sequence—one with the G59C substitution and the other with L83Y. Figure 7C above shows the plotted Root Mean Square Deviation (RMSD) value of the protein complexes, modelled over a 50 ns timescale. Reflecting the SIFT score and DynaMut2 result, RBCS bearing the L83Y substitution negatively impacted stability, resulting in a high degree of variability in the RMSD value before plateauing to approximately 3.9 Å at 46 ns. Low stability (relative to the other modelled protein complexes) was additionally observed for the RuBisCO complex containing the *O. sativa* RBCS, plateauing to approximately 3.98 Å at 34 ns. The wildtype RuBisCO complex derived from *H. vulgare* exhibited similar stability to the complex containing *O. sativa*-derived RBCS, except with fewer fluctuations to the RMSD value over the 50 ns timescale, plateauing to approximately 3.93 Å at the 27 ns timepoint. Modelling of the wildtype *Z. mays* RBCS complexed with *H. vulgare* RBCL, revealed a comparatively lower RMSD value throughout the simulation, maintaining an average value of 2.89 Å after 30 ns. Interestingly, modelling the *H. vulgare* RuBisCO complex containing the mays-derived G59C mutation revealed comparable stability to the complex containing the wildtype maize RBCS, with an average value of 2.97 Å after 30 ns. Figure 7D, shows the G59C substitution in the βA-βB loop of the small subunit, as generated by DynaMut2. The output showed that a hydrogen bond forms between the βA-βB loop and a large subunit of RBCL, which may substantiate the observed increase in stability for *H. vulgare*-derived RBCS bearing the G59C mutation. The combined results of molecular dynamics analysis indicate that C3-derived RBCS generally imparted lower stability on the rubisco complex versus the sequence of C4 RBCS and highlights Cysteine at position 59 as a potential contributor to enhanced RuBisCO stability.

### 2.7. Natural Selection Analysis Reveals Distinct Selection Pressures in the RBCS Gene Family Lineages of Distinct Phylogenetic Groupings

Phylogenetic analysis by maximum likelihood (PAML) was performed to detect selection pressures across RBCS lineages. First, we aimed to elucidate if there was any specific selection pressure towards residues in the panicoid (C4) RBCS branch versus the other grasses in this study. Second, we aimed to determine if there was increased selection pressure on the Chr5 RBCS lineage compared to the Chr2 RBCS lineage in the *Triticeae*. In turn, four phylogeny groups were compared—C4 RBCS (ω_C4_), Chr5 RBCS in the *Triticeae* (ω_Ch5_), Chr2 RBCS in the *Triticeae* (ω_Ch2_) and all other grass RBCS sequences (ω_rest_). Branch-specific tests were performed to identify the *dN/dS* ratio (ω) (ratio of synonymous to non-synonymous mutations) of select phylogeny groups under differing assumptions. As shown in Table 1, under the one ratio model (which assumes ω is the same for all phylogeny groups), the *dN/dS* ratio was 0.04908. When we expanded the model to two branches, ω (*dN/dS*) was 0.0479 (ω_Ch5_ = ω_Ch2_ = ω_rest_) and 0.1999 (ω_C4_), indicating that despite relatively strong negative selection observed for both branches, there is more relaxed selection pressure in the C4 RBCS lineage. Under the three-branch model, we compared relative selection pressures for *Triticeae* RBCS on Chr5 and *Triticeae* RBCS on Chr2. From the results of the 3-branch model, it was deduced that more stringent selection pressure is applied to the chr2 RBCS branch (ω = 0.02328) versus the chr5/1 branch (ω = 0.40993). This is interesting given that the chr5 copies are the most highly expressed in the *Triticeae*, as observed for rye, Aegilops and barley (Figure 5). This result could indicate that the tandem duplication array in *Triticeae* species serves a specific accessory function to photosynthesis, despite comparatively lower expression levels, as evidenced by the increased stringency on amino acid sequence conservation. Finally, we completed a 4-branch model analysis, where all four phylogeny groups were considered under different selection pressures. Under the four-branch model, ω (*dN/dS*) was 0.04755 (ω_rest_), 0.20165 (ω_C4_), 0.02307 (ω_Ch2_), and 0.44535 (ω_Ch5_) (Table 1).

Next, we further investigated the C4 RBCS, Chr2 *Triticeae* RBCS, and Chr5 *Triticeae* RBCS lineages to identify residues that may be under positive selection. Likelihood ratio tests (LRTs) were performed to determine the best model for running subsequent site and branch models in downstream analysis. Comparing the one-branch to the two-branch model and the two-branch to the four-branch model, the two-branch model (with C4 RBCS sequences as the foreground lineage) was determined to be the best-fit model for analysis, with a *p*-value of 0.0041. The results of the LRTs thus indicated that the C4 lineage is subjected to a different selection pressure relative to the C3 lineages (i.e., ω_C4_ ≠ ω_Ch5_ = ω_Ch2_ = ω_rest_). Site-branch models were used to investigate whether the C4, Chr2 and Chr5 groups were under positive selection. For the Chr2 group RBCS sequences, the branch and site-specific model revealed no residues under positive selection (Appendix A). In contrast, the model results for the Chr5 group *Triticaeae* RBCS sequences revealed that residue 103A was under positive selection, with a probability of 0.999 (Table 1). Interestingly, residue 103A was only specific to the chromosome 5 group RBCS and relatives, with all chromosome 2 copies possessing serine at this position in select *Triticaeae* members, including *H. vulgare*, *S. cereale*, *A. tauschii*, and *T. urartu.* This may suggest a divergent function between the two groups, potentially explaining the differential expression patterns observed in upstream analysis (Section 2.4). Site-branch test results for the C4 group RBCS sequences, revealed a large number of sites under selection, including 70T (in the αA helice), and 102N, 103A, and 111R, which are located in the catalytically relevant βA-βB loop. Another residue of interest was 143A, which was within βC in the βC-βD loop. Interestingly, this residue is conserved across 100% of C3 representatives in this study, but with high diversity in the C4 clade. *O. thomaeum* has C at this position, *P. hallii* H or N, *U. fusca* C and *S. italica* A or C. The final step of natural selection analysis was the completion of LRTs to determine if ModelA was significantly better than its null hypothesis, thus indicating the identified amino acid sites are indeed under selection. As expected, the LRT was insignificant for the Chr2 group, where no sites were identified as under selection (*p* = 1.00). LRTs for the Chr5 group RBCS and C4 RBCS branches were significant, with *p* values of 0.0367 and *p* < 0.0001, respectively. The above analysis indicated that C4 group RBCS sequences may be under greater evolutionary change than other C3 grass RBCS sequences, which, in turn, allows for the greater adaptability of C4 rubisco complexes over time. In addition, several residues in the βA-βB loop were identified as under selection in the C4 group and Ch5 group RBCS sequences, which indicated that catalytically pivotal regions were targeted by positive selection. Finally, these observations may suggest that the divergence in sequence at position 103 between Chr2 and Chr5 copies may serve a specific functional purpose in the RBCS gene family possessed by *Triticeae* members.

## 3. Discussion

### 3.1. Expanded RBCS Arrays in the Triticeae/Aveneae May Compensate for Reduced Catalytic Efficacy and/or Enhance Adaptability to Temperate Environments

In this study, we observed large variability in *RBCS* gene family size across grass lineages, with a greatly expanded *RBCS* family observed in members of the *Triticeae*/*Aveneae* (as covered in Section 2.1). Genome expansion/duplication in the *Triticeae* is a well-documented phenomenon. *Triticeae* members possess large and highly dynamic genomes compared to their relatives, consisting of over 80% transposable element sequences which assist in continuous modification through time [26,27]. Given the high degree of correlation between genome size and *RBCS* copy number (R = 0.81, *p* < 0.001), genome duplication event(s) in part likely contributed to expanded *RBCS* copy number in representatives of the *Triticeae*/*Aveneae*. We deduced that this occurred prior to the split of these lineages in a common ancestral species due to the high degree of synteny observed between *RBCS* copies on all chromosomes in *H. vulgare*, *T. urartu*, *A. tauschii*, *S. cereale* and *A. atlantica* (Figure 3B–D).

The proximity of chromosome 2 genes also supports the hypothesis that all chromosome 2 *RBCS* expansions in the aforementioned species may have resulted from a tandem duplication event. Specifically, the location of *RBCS* gene(s) at the terminal end of chromosome 2 (chromosome 3 in *A. atlantica*) indicates that sub-telomeric duplication events may have facilitated a more rapid expansion of the gene family, thereby enhancing the environmental adaptation of photosynthetic efficiency [28]. A study by Hanada et al. [29] found that the nature of gene duplication events in *Arabidopsis* reflected the adaptive patterns of the duplicate gene copies. Tandem duplications were associated with environmental stress response, while other duplication types correlated more with intracellular regulation [29]. Supporting this notion, a high proportion of cis-regulatory elements linked to drought, stress, and light conditions was observed in the promoters of the grass species studied, with light and drought response elements comprising over 30% of CRE composition across all lineages (and up to 59% in some instances) (Figure 4A). The authors also noted that the tandem expansion of gene families was lineage-specific, attributed to changes in distinct species that reflect their unique environmental exposures [29]. Given that members of *Triticeae* examined in this study are generally adapted to temperate regions, we suggest that the observed tandem expansions in *H. vulgare*, *T. urartu*, *A. tauschii*, *S. Cereale*, and *A. atlantica* likely occurred independently. This is likely due to an increased need to upregulate photosynthesis, caused by the lower sunlight intensity seen at 40–60° latitudes [30,31]. Furthermore, all *RBCS* copies retained an expression profile in *H. vulgare*, *S. Cereale*, and *A. Tauschii* (Figure 5) and were also retained in the expression of all members of the barley pan-transcriptome (Appendix A). The absence of pseudogene development across lineages suggests an adaptive functionality of the expanded array, potentially to enhance photosynthesis in temperate environments or to better regulate photosynthetic response through differential expression mechanisms [32,33].

Finally, we observed higher expression in winter-type barley pangenome accessions for both the coleoptile and shoot (primary regions where photosynthetic processes occur). This further highlighted the association between plants adapted to cool regions that modulate their photosynthetic response to maximise biomass generation under low sunlight intensities (Figure 6). This trend is echoed in other studies, such as that of Huner et al. [34], which reported that winter-adapted plant cultivars exhibit higher efficacy of the photosystem apparatus and greater resistance of photosynthetic processes to reduced light intensities. Another study demonstrated the high yield capacity of the winter wheat cultivar Weimai 8, resulting from superior performance of photosystem II compared to the control [35]. Despite the positive associations reported between winter adapted crops and photosynthetic traits, the impact of enhanced *RBCS* expression observed in the winter type barley pangenome accessions remains elusive without further investigation. It is widely known that winter-type barley accessions generally exhibit higher yield stability than their spring-type counterparts, which shows promise for an association of cold adaptation with photosynthetic vigour [36]. However, such studies primarily attribute phenology differences (i.e., earlier maturing of winter type cultivars) to superior winter-type biomass accumulation, and improved yield stability [36,37]. Hence, a thorough assessment is needed of relative RuBisCO catalytic efficacy in spring- and winter-type varieties and their correlation with RBCS expression levels and biomass accumulation. The findings may then better elucidate a relationship between cool climate adaptation and photosynthetic efficiency. A final consideration is the relative sensitivity of winter-type varieties to climatic conditions, and the potential of other environmental triggers (e.g., low temperature and/or low light induced gene pathways) affecting winter-type *RBCS* expression [38]. Overall, the combined phylogenetic and transcriptome-based observations of the *Triticeae* in our study suggest a photosynthetic apparatus temporally adapted to cool climatic conditions through tandem expansion and modulation of expression. High winter-type *RBCS* expression in the barley pan-transcriptome is compelling, yet we recognise the potential caveats of small sample size in the winter-type group, which may conflate observed trends [39]. The identification of regulatory mechanisms surrounding enhanced winter-type *RBCS* expression would be highly beneficial. Such findings may be adapted to spring-type barley genotypes, which have been identified to perform more robustly in a future shaped by climate change [40].

### 3.2. Exploring the Terminal I Box Element Pair as a Potential Enhancer of Rubisco Expression and Modulator of Photosynthetic Rate

Light-induced I-box elements are a well-known feature of *RBCS* promoters, with numerous studies documenting their effects on *RBCS* expression. Historical studies involving tomato (*Solanum lycopersicum*) and *Arabidopsis* have shown that I-box elements are required to maintain high levels of *RBCS* expression [41,42]. Furthermore, cotton *RBCS* promoters containing I-box elements exhibit constitutive expression levels similar to those of the Cauliflower mosaic virus (CaMV) [43]. However, a more recent study found that native wheat *RBCS* promoters are less efficient than CaMV promoters based on GUS reporter gene expression. This may indicate that other factors, such as species-specific promoter CRE arrangements, may also influence the effectiveness of *RBCS* expression [44]. Our study observed that all highly expressed *RBCS* gene copies in representatives of the *Triticeae* possessed a terminal I-box element pair, separated by a spacer sequence approximately 26 bp in length in most sequences (Figure 5). Thus, there is potential that the spatial arrangement and frequency of I-box elements at the terminal region of *RBCS* promoters may greatly enhance *RBCS* expression and, consequently, the rates of photosynthesis in these gene copies. Studies in tomato have shown that the spatial arrangement of I and G box elements is critical for maintaining gene expression [42]. Additionally, the characterisation of conserved modular array 5 (CMA5) in *RBCS8B* (derived from *Nicotiana plumbaginifolia*) showed that the expression of some *RBCS* family members relies on the presence of specific light-reactive CREs (in this case, an I- and G-box arrangement) [45]. In fact, one study demonstrated that the distance between the I and G box elements in CMA5 is critical for ensuring adequate strength of the transcription signal [46]. We know that RuBisCO large subunit expression is influenced by the relative transcript levels of *RBCS*, and thus *RBCS* expression somewhat directs the rate of photosynthesis. The identified terminal I-box element arrangement in this study may thus serve as a pathway to modify promoter composition of accessory *RBCS* copies on chromosomes 2/3 in barley and rye [47,48]. Such modifications may enhance global *RBCS* expression, subsequently improving photosynthetic efficacy and biomass generation [49]. Overall, the terminal I-box element pair arrangement may present a promising avenue for enhancing photosynthesis in cereal crop species of nutritional relevance. However, given that our current evidence for I-box induced expression is computational, we establish this inference in a cautionary manner. As such, before targeted modifications can occur, functional characterisation experiments will need to be completed to confirm that the terminal I-box pair does indeed impart high *RBCS* expression levels. Validation of I-box associated enhancement of *RBCS* expression via CRISPR-induced removal of the terminal I-box pair would serve an appropriate validation approach [50].

### 3.3. Native C4 Polymorphisms in the βA-βB Loop Serve as Novel Polymorphisms for the Targeted Improvement of C3 Rubisco Complexes

An assessment of the C4 grass clade reveals a somewhat contrasting pattern in terms of copy number. Given the current understanding of C4 type RuBisCOs and their enhanced catalytic efficacy, C4 grasses may not have necessitated the development of a large *RBCS* gene family to support photosynthetic processes [51]. In this study, the observed capacity for enhanced RuBisCO complex stabilisation in C4 RBCS was demonstrated through in-silico molecular dynamics. Following the in-silico analysis, we found that wildtype RBCS derived from the C4 crop *Z. mays* provided the greatest stability (Figure 7). These observations somewhat echo previously reported results from hybrid RuBisCO experiments. Matsumura et al. [20] demonstrated that rice RBCL complexed with RBCS derived from *S. bicolor* exhibited superior catalytic performance compared to the wildtype RuBisCO complex, likely attributable to specific residues in the βC-βD hairpin of *S. bicolor* RBCS. This research further supports the idea that RBCS, despite lacking any catalytic components, is adapted to influence the overall quality of the RuBisCO complex. In our study, we identified a C4-specific amino acid substitution, G59C, in the βA-βB loop region of RBCS, which significantly enhanced the capacity of barley RBCS to stabilise the RuBisCO complex (Figure 7). The βA-βB loop is recognised as a site of catalytic importance in RBCS. Firstly, the length of the βA-βB loop is believed to regulate the size and width of the solvent channel [14,52]. In addition, mutations of residues present in the loop region, especially those near RBCL, influence the kinetic activity of RuBisCO, particularly its specificity for the CO_2_ substrate [53]. Our results from PAML analysis also highlighted several residues in the βA-βB loop that were under positive selection in the C4 lineage, emphasising the loop region’s critical role in the evolution of RuBisCO catalytic efficiency (Table 1). Among the residues under positive selection were 102N and 103A (55N and 56A based on amino acid sequence position in *S. oleracea*, respectively), which were only three and two residues away, respectively, from the cysteine residue ‘59C’ naturally occurring in *Z. mays* RBCS (Table 1; Figure 7). Finally, Dynamut2 revealed that the G59C substitution leads to the formation of a hydrogen bond with Lysine at position 161 in the large subunit of *H. vulgare* (Figure 7D), likely contributing to enhanced stability of the assembled RuBisCO complex [54]. Given these observations, we have identified the G59C substitution in the βA-βB loop region as a potential candidate for improving the catalytic potential of C3 crop RBCS. However, given the current lack of experimental validation of this novel protein substitution, our inferences remain conservative at this stage. Future functional analyses will be required to understand the true impact of the G59C substitution on the assembled rubisco complex. Potential validation protocols include differential scanning calorimetry (DSC) to assess reactive protein stability following thermal exposure. In addition, investigation of enzyme efficiency (*K*_cat_) may be completed following CRISPR-induced knockout of the *RBCS* multigene family in barley and subsequent assembly of a modified barley rubisco complex comprising wildtype *HvRBCL* and *HvRBCS* possessing the novel G59C substitution [20,55].

### 3.4. Specific Selection Pressures in the Expanded RBCS Gene Family Are Suggestive of Potential Adaptive Divergence of RBCS Copies in the Triticeae

Adaptive divergence in the *RBCS* gene family has previously been observed in other species [56,57,58]. For example, authors have found that rice *OsRBCS1*, located on Chromosome 2, exhibits expression in the root, anther, culm and leaf sheath but no expression in the leaf blade (the primary site of photosynthesis) [59]. Given the lower CO_2_ specificity of *OsRBCS1* compared to its gene family relatives, it was proposed that *OsRBCS1* expression may produce a more efficient RuBisCO complex in regions like the leaf sheath, which exhibit higher CO_2_ concentrations [59]. In our study, it was found that selection pressures were prevalent upon comparison of Chr5/1 and Chrs2 *RBCS* lineages in the *Triticeae*, with specific selection pressure identified for residue 103A in the *HvRBCS5* reference sequence (Table 1). Given that all Chr5 RBCS copies possess the A residue at this position (in contrast to Chr2 RBCS possessing residue S), this could indicate a specific evolutionary pressure for A residue maintenance at position 103. In addition, given the observed high expression of the Chr5 *RBCS* gene copies in *S. cereale*, *H. vulgare*, and *A. tauschii* in this study, there exists a potential evolutionary favouritism of Chr5 *RBCS* functionality in the *Triticeae*. This may have led to observed shifts in expression such that Chr5 copies express the highest by a substantial degree. This pattern of favoured expression of specific *RBCS* copies echoes across various plant lineages, including *A. thaliana*, *O. sativa*, *S. lycopersicum* and *S. tubersum* [14]. However, little understanding can be gained as to why specific *RBCS* gene copies are preferentially expressed over others—especially given the high sequence homology observed between them [60]. One explanation for the combined low expression of tandem array copies in *Triticaeae* may be that they serve to supplement the full photosynthetic effect or otherwise may assist in the fine-tuning of environmentally sensitive responses [14,61]. For example, the expanded *RBCS* Chr2 copies observed in the *Triticeae* may ramp up more under specific stress-induced pressures to assist in photosynthetic efficacy—a notion further substantiated by our observation of unique promoter composition across gene family members at the species level (Figure 4A).

Another study that investigated the sequence composition of ancestral plant RuBisCOs noted the difficulty in isolating key RBCS residues that may improve catalytic efficiency. Authors observed unique residues reported in an efficient potato RBCS were not present in any of the high *K*_cat_ ancestral RBCS sequences [62,63]. In turn, this highlights that adaptation of RBCS catalysis across species results from multiple instances of differing sequence composition, due to temporal exposure to varied environments. Hence, diversity in RBCS evolution across species may provide a rich source of potential for future modification and catalytic fine-tuning of rubiscos in commercial crops. Ultimately, further work is needed to gain a better understanding of the *RBCS* gene family’s function across species and the roles of specific *RBCS* sequences in photosynthesis and beyond.

## 4. Materials and Methods

### 4.1. Phylogenetic Characterisation of the RBCS Gene Family and Assessment of Copy Number Variation Across True Grass Genera

A total of 17 plant species were selected for analysis, including 16 members of the true grasses (family *Poaceae*) and *Arabidopsis thaliana* as the outgroup species. All selected species possessed a diploid genome arrangement to deconvolute any potential associations of observed *RBCS* copy number with ploidy level for any given species. Genome assemblies were retrieved from various sources, summarised in Table 2 below. The study incorporated true grass representatives from both C3 (*n* = 10) and C4 (*n* = 6) metabolising classes, including basal members from both the BOP (subfamilies *Bamusoideae*, *Pharoideae*) and PACMAD clades (subfamily *Chloridoideae*).

OrthoFinder version 2.5.5 was used to identify RBCS orthologues through homology-based search of species amino acid sequences. Identified orthologues were then further screened with conserved domain batch search (https://www.ncbi.nlm.nih.gov/Structure/bwrpsb/bwrpsb.cgi, accessed on 29 April 2023) for the presence of RBCS-associated domains, namely pfam12338 and pfam00101. Any severely truncated orthologues with insufficient domain architecture were subsequently removed before any downstream analysis. Following the filtering step, identified *RBCS* sequences were assessed for phylogenetic relationships between species lineages. First, an amino acid alignment of RBCS sequences (*n* = 89) was generated with ClustalO (https://www.ebi.ac.uk/jdispatcher/msa/clustalo, accessed on 5 May 2023) with an MBED-like clustering iteration, and default parameter settings for combined number of iterations (0), max HMM iterations, and max guide tree iterations. The alignment was then passed to raxmlGUI version 2.0 for phylogenetic tree construction, using the JTT amino acid model with ML + rapid bootstrap analysis and gamma substitution rates for 100 repetitions [66]. The resulting tree file was then ported to FigTree version 1.4.4 for subsequent gene label annotation and production of the finalised tree graphic. Correlation analysis of RBCS copy number versus genome size was carried out using the Pearson correlation method as part of the ggpubr R software package (version 0.6.0). Protein FASTA files and data frames used for analysis (including the master file containing all gene identifiers and their gene names), are available in Appendix A.

### 4.2. Investigation of Relative Copy Number of the RBCS Gene Family in the Expanded Barley Pangenome

The genome assemblies of 71 newly sequenced barley (*H. vulgare*) pangenome varieties were obtained from Jayakodi et al. [67]. The Gene Model Mapper (GeMoMa) pipeline version 1.9 was selected to run projections for newly sequenced pangenome accessions, to generate gene annotations for use in downstream analysis [68]. Projections were run remotely on the Setonix supercomputer using a research allocation provided by the Pawsey Supercomputing Research Centre in Kensington, Western Australia. Pangenome annotations generated as part of this project can be found in Appendix A. FASTA files of protein sequences for each barley accession were then generated with gffread version 0.12.7 in preparation for sequence homology search [69]. Using the Many-against-Many sequence searching (MMseqs2) software suite (release version 15-6f452), a combined protein database derived from the barley pangenome varieties was searched using the *RBCS* gene copy HORVU.MOREX.r3.1HG0037500.1 from the MorexV3 reference sequence as the query, with thresholds of 90% identity and 80% coverage [26,70]. Copy number counts were then assimilated from the MMseq2 sequence hits and polar bar plots generated with the count data using the ggplot2 version 3.5.1 package in RStudio (version 4.4.1).

### 4.3. Synteny Analysis of the Expanded RBCS Gene Family in the Triticaeae/Aveneae

The JCVI toolkit python library (https://github.com/tanghaibao/jcvi, accessed on 18 June 2023) was used to identify syntenic blocks comprising RBCS gene family copies, shared between members of the *Triticeae* and *Aveneae* tribe [71]. Macrosynteny and microsynteny plots were then generated with JCVI, highlighting syntenic *RBCS* genes identified on chromosomes 1, 2 and 5, respectively, and using the barley (*H. vulgare*) chromosome as the reference for mapping. 

### 4.4. Extraction of Upstream RBCS Gene Regions and Identification of Cis-Regulatory Elements in the True Grasses and the Barley Pangenome

The sequence region located 2000 bp upstream of the gene start site was extracted from the genome assemblies of both the true grass family members and the barley pangenome accessions using BEDtools version 2.31.0. Extracted upstream regions were then submitted to the PlantCARE online database (https://bioinformatics.psb.ugent.be/webtools/plantcare/html/, accessed on 20 September 2023) to determine the cis-regulatory element (CRE) profile of the query sequences [72]. Identified promoter elements were then filtered for CREs commonly associated with the regulation of rubisco small subunit expression using information derived from a combined literature search, yielding six CREs of interest [41,44,73]. Promoter element maps of the filtered results were then created with TB tools (version 2.0) using the BioSequence structure illustrator [74]. Input files required to generate the promoter element map included a list of the upstream *RBCS* promoter sequences and their lengths (2000 bp), a four-column ‘Input ID list’ with Gene code, motif start position, motif end position and motif name, and phylogenetic tree of *RBCS* sequences (Newick format). A presence-absence matrix was additionally created for true grass family species via processing of the unfiltered PlantCARE output with pandas (version 2.0.3) and plotted with ggplot2. Plotted matrix CREs were colourised based on the CRE function. Pie charts depicting relative percentages for each CRE function type were additionally generated using ggplot2.

### 4.5. Investigation of RBCS Gene Family Copy Relative Expression Observed in H. vulgare, A. tauschii and S. cereale

RNA sequencing (RNAseq) data derived from *H. vulgare*, *A. tauschii* and *S. cereale*, was retrieved to determine which gene copies exhibit the greatest levels of *RBCS* transcript expression. Expression data for *H. vulgare* was retrieved from the BarleyExpDB (http://barleyexp.com/index.html, accessed on 15 September 2023) using Bioproject PRJEB14349: RNA-seq of 16 developmental stages of barley—and taking epidermis tissue expression (EPI) only [75]. RNAseq data for *A. tauschii* was obtained from Bioproject accession PRJNA748580 and treatment run SRR15206207 (https://www.ncbi.nlm.nih.gov/sra/SRX11512710[accn], accessed on 15 September 2023), pertaining to wheat rust resistant *Aegilops tauschii* that had not been inoculated with the rust pathogen [76]. *S. cereale* RNAseq data was retrieved from the European Nucleotide Archive (ENA), under experiment accession ERX3662813 and run number ERR3671408 (https://www.ebi.ac.uk/ena/browser/view/ERR3671408, accessed on 15 September 2023). For *H. vulgare* expression data, RNAseq results were provided directly as Fragments Per Kilobase Per Million mapped fragments (FPKM) following a database search of MorexV3 assembly RBCS gene queries. Further processing of raw RNAseq values (provided in paired layout), was required for *A. tauschii* and *S. cereale*, and this was achieved using Kallisto software version 0.5.0 to generate estimated transcript abundance values (as transcripts per million, or TPM) for each species [77]. The relative transcript abundance was then calculated as a percentage (%) for each species and plotted against a phylogenetic tree of the *RBCS* sequences and CRE map (see Section 4.4 for how CRE maps were generated). The treeio (version 1.28.0), ggtree (version 3.12.0), and aplot (version 0.2.3) packages were used for plot creation in RStudio version 4.4.1 [78,79,80].

### 4.6. Assessment of Relative RBCS Expression in the Barley Pan-Transcriptome

Relative transcript abundance was compared for *RBCS* gene copies obtained from novel transcriptome data derived from 20 barley accessions comprising the original v1.0 barley pangenome collection and provided by the Western Crop Genetics Alliance for analysis [81]. Raw data reads can be accessed from the European Nucleotide Archive (ENA), under bioproject accessions PRJEB64639 and PRJEB64637 (https://www.ebi.ac.uk/ena/browser/search, accessed on 20 November 2023). Before the interrogation of genotype-specific reference transcript datasets (GsRTDs), homologous *RBCS* gene copies were first identified from the pan-transcriptome annotations via blastp of the MorexV3 RBCS protein sequences (HORVU.MOREX.r3.5HG0468970, HORVU.MOREX.r3.5HG0469020) against protein databases derived from the pan-transcriptome accession annotation data. Transcript abundance (TPM) was calculated using the sum of raw TPM data values across the *RBCS* gene families of pangenome varieties possessing a spring-type growth habit (*n* = 14) and winter-type growth habit (*n* = 5) and plotted as a simple bar chart with ggplot2. TPM values for each variety are provided in Appendix A. Student *t*-tests were then performed for the distribution of summed raw TPM value(s) across all *RBCS* gene copies, between the spring and winter type varieties, for three biological replicates collected from five tissue types—Inflorescence (In), Coleoptile (Co), Caryopsis (Ca), Root (Ro), and Shoot (Sh) [81].

### 4.7. Molecular Dynamics Analysis of in Silico Rubisco Hybrid Complexes

Before molecular dynamics (MD) analysis, a combination of methods was used to select amino acid substitutions for in silico mutagenesis in the RBCS of *H. vulgare.* First, an amino acid alignment (generated with ClustalO: see Section 4.1) for all true grass RBCS included in this study, was interrogated using SnapGene software version 7.0 to identify dominant amino acid residues in the C3 photosynthesising grasses and the C4 photosynthesising grasses. Any residues exhibiting substantial amino acid divergence between the C3 and C4 grasses (for example, being present in at least half of the C4 group species and not present in any representatives of the C3 group at the same position) were considered potential candidates for mutagenesis. The experimentally verified protein structure file (PDB: 6KYI) of wildtype *O. sativa* generated by Matsumura et al. [20] was then retrieved from the Protein Data Bank (https://www.rcsb.org/search, accessed on 25 September 2023) and imported to PyMOL software (version 2.5.5). PyMOL was used to identify amino acid residues in the RBCS chains within 4 angstrom (Å) units of the large subunit molecule(s) that may influence RBCL catalysis. Residues identified near the large subunit and homologous between the C3 species *H. vulgare* and *O. sativa* (but differed in *Z. mays*) at the same position were considered for upcoming interrogation with sequence-based predictor software. The next step involved the use of the Sorting Intolerant From Tolerant (SIFT) engine (https://sift.bii.a-star.edu.sg/www/SIFT_seq_submit2.html, accessed on 29 September 2023) to query the predicted impact of selected substitutions of interest in the protein sequence of HORVU.MOREX.r3.5HG0469020.1. SIFT parameters included a median sequence conservation of 3.00, and the selected database was the UniProt-SwissProt 2010_09. The DynaMut2 structural-based predictor (https://biosig.lab.uq.edu.au/dynamut2/, accessed on 29 September 2023) was also used to discern amino acid substitution impacts on protein function, using an *H. vulgare* wildtype rubisco complex Protein Data Bank (PDB) file generated with AlphaFold2 (ColabFold version 1.5.5) and using HORVU.MOREX.r3.5HG0469020.1 (HvRBCS1) as the RBCS amino acid backbone sequence [82].

Additional hybrid protein PDB files were created with AlphaFold2, including *H. vulgare* RBCL complexed with *Z. mays* RBCS1 (Zm00001d052595) and *H. vulgare* RBCL complexed with *O. sativa* RBCS3 (Os12g0291100). AlphaFold2 parameters included selection of the alphafold2_multimer_v3 model (generally selected for modelling of complex multimers), utilising a greedy pairing strategy for enhanced prediction of complex protein structure. The model was run with 3 recycles, and a maximum of 200 iterations. Based on outcomes of the combined RBCS protein alignment, sequence-based predictor and structure-based predictor analyses, two in silico *H. vulgare* rubisco mutants (one possessing the L83Y substitution in RBCS, and the other possessing the G59C substitution in RBCS, using the spinach (*S. oleracea*) RBCS as the reference sequence for amino acid position) were generated using the PyMOL mutagenesis wizard. Substitutions involving a change from an aliphatic (L) to aromatic (Y) residue, may substantially alter protein properties, such as stability, in part due to the increased hydrophobicity imparted by the aromatic ring structure [83]. L83Y was thus considered an intriguing target for modelling analysis, combined with the knowledge that L83 is a highly conserved residue across plant representatives [14]. The G59C substitution was primarily considered an appropriate candidate for modelling, given its prevalence among C4 grass in our study, and its location in the βA-βB loop region (a critical region known to impact rubisco catalysis) [53]. All PDB files were then utilised in the subsequent molecular dynamics simulation(s) using GROMACS version 2023.3. RuBisCO protein complexes were modelled utilising the OPLS-AA/L (all-atom optimised potentials for liquid simulations (long chain)) forcefield and the SPC/E (extended simple point charge) 3-point water model, which implements the ideal shape of water molecules (109.47 degrees) [84]. The selection of the OPLS-AA/L forcefield was justified based on its suitability for large biomolecule modelling, and its excellent balance of accuracy and efficiency, which was important for a moderate simulation time [85]. Methods utilised for MD analysis were adapted based on the protocols developed by Lemkul (2019) [86]. First, imported PDBs were stripped of crystal water molecules, and the parameters of a simple cubic box were defined as the system for MD simulation, ensuring a minimum distance of 1 nm between the modelled protein and box edge (to satisfy periodic boundary conditions). The system was then solvated with water, and any net charge present in the protein solute was balanced by adding ions to generate an electroneutral system suitable for subsequent energy minimisation. Systems producing negative potential energy with an order of approximately 10^5^–10^6^ and a maximum force not exceeding 1000 kJ mol^−1^ nm^−1^ were determined not to possess abnormal/interfering atomic arrangements (steric clashes) and were thus suitable for equilibration temperature and pressure. Equilibration of temperature (utilising a reference temperature of 300 K) and pressure (utilising a reference pressure and 1 bar) was completed for 100 picoseconds (ps) while imparting position restraints on system-heavy atoms to allow for system stabilisation without substantial structural changes to the protein [84]. Equilibration was considered successful following the observed plateau of temperature fluctuations around the 300 K reference value and an observed stable density fluctuation sustained around 1008 kg m^−3^ (the expected density of an SPC/E modelled system) [86]. Finally, MD runs were completed for all equilibrated systems over a 50 nanosecond (ns) timescale, a common simulation length for complex protein modelling [87]. Parameters specified for the MD runs included implementation of a modified Berendsen theromostat (tcoupl = V-rescale) combined with rapid thermal coupling parameters (tau_t = 0.1 ps) and independent equilibration of temperature baths for protein and non-protein regions of the system, to assist with artefact reduction (tc-grps = Protein Non-Protein). Pressure related controls included use of the Parrinello-Rahman barostat combined with slowed pressure coupling (tau_p = 2.0 ps). H-bonds were constrained in our system and subsequently accommodated by the LINCs algorithm and 2 femtosecond timestep (dt = 0.002). Simulations were run concurrently on five Nimbus cloud compute instances, each comprising 16 VCPUs and 64 GB of RAM, utilising an allocation provided by the Pawsey Supercomputing Research Centre.

Following dynamics, computation trajectories were corrected to compensate for spurious protein behaviour resulting from system periodic boundary conditions [86]. The relative stability of the modelled protein complexes was then compared by plotting the root mean square deviation (RMSD) values (Å) over the 50 ns timescale.

### 4.8. Natural Selection Analysis of RBCS Sequences in the True Grasses

Natural selection analysis was performed with Phylogenetic Analysis by Maximum Likelihood (PAML) software (version 4.9) utilising an alignment of RBCS CDS sequences generated with MEGA (version 11), and using HORVU.MOREX.r3.5HG0468970 (HvRBCS5) from *H. vulgare* as the reference sequence [88,89]. Analysis was performed on four phylogeny groups—RBCS sequences from C4 grasses (ω_C4_), sequences comprising chromosome 5 RBCS in the Triticeae (ω_Ch5_), sequences comprising chromosome 2 RBCS in the Triticeae (ω_Ch2_) and all other grass species RBCS sequences grouped as ω_rest_. Branch-specific models revealed the ratio of non-synonymous to synonymous amino acid substitutions (omega (dN/dS)) for specified phylogenetic protein groupings. Specification of branch-specific model parameters included the one-branch model—assuming ω is the same for all branches (ω_C4_ = ω_Ch5_ = ω_Ch2_ = ω_rest_), two-branch model—assuming ω (C4 branch RBCS sequences) does not the equal ω of all other the RBCS sequences studied (C3 members) (ω_C4_ ≠ ω_Ch5_ = ω_Ch2_ = ω_rest_), the 3-branch model, assuming that ω (Chr5 and Chr2 *Triticeae* RBCS sequences) does not equal ω of the other phylogeny groups (ω_Ch5_ ≠ ω_C4_ = ω_rest_ ≠ ω_Ch2_) and the four-branch model which compared all four phylogenetic groups (assumed ω_C4_ ≠ ω_Ch5_ ≠ ω_Ch2_ ≠ ω_rest_). Codeml parameters for all branch-specific models included implementation of codon analysis (Seqtype = 1) utilising a standard genetic code conversion table (icode = 0), and the F3X4 model (CodonFreq = 2), which takes into account codon position bias, thus improving the accuracy of dN/dS estimates. The model = 0 parameter was set for the one-branch model, and the model = 2 parameter was set for models assessing two or more branches. Files used in the analysis, including code parameter files, are in Appendix A. For each set of model analyses, a phylogenetic tree (with branches pertinent to analysis labelled) was provided, along with an alignment file of the RBCS sequences.

Likelihood ratio tests (LRTs) were then completed to determine the most suitable branch model based on the value of 2Δ*L*. First, we derived the value of 2Δ*L*, utilising the following equation:2Δ*L* = 2(lnL[model1] − lnL[model2])

The *p*-value was then determined from both 2Δ*L* and the DF value, whereDF = np [model1] − np [model2]

The site and branch-specific tests were carried out with two specified foreground lineages (C4 branch RBCS and Chr5 *Triticaeae* RBCS) to identify potential residues under positive selection pressure. Two separate models were ran and compared for the respective foreground lineages. Model A assumed positive selection pressure in foreground lineage(s), and conserved or neutral selection in background lineage(s) (model = 2, NSites = 2, fix_omega = 0, omega = 1.5). Model A null assumed conserved or neutral selection pressure for all lineage(s), fixing omega (ω) to a value of 1 (model = 2, Nsites = 2, fix_omega = 1, omega = 1). LRTs were subsequently performed to compare model A and model A null, to determine whether residues were in fact subjected to positive selection pressure based on the statistical significance threshold (*p* < 0.05). The site and branch-specific LRTs performed here were carried out utilising the same protocol listed above for the calculation of the 2Δ*L* and the DF value. The above protocol was adapted from the methodology developed by Jia et al. [25].

## 5. Conclusions

Our study revealed diverse *RBCS* gene family compositions across various true grass species. The *Triticeae* tribe exhibited a significant expansion of the gene family, which greatly contrasted with the C4 members *Z. mays* and *S. bicolor* (which historically demonstrate greater catalytic efficiency of the rubisco complex). Extensive synteny of the *RBCS* gene family was observed in the *Triticeae*, with syntenic relationships observed spanning multiple chromosomes. We identified chromosome 2 *RBCS* as the likely ancestral copy, given the syntenic relationships shared between both *Triticeae* and *Aveneae RBCS* sequences. The promoter composition of the *RBCS* gene family showcased predominant light and drought-responsive elements, followed by other stress-associated CREs, reflecting the primary function of RBCS in photosynthetic processes. Furthermore, it suggests that rubisco expression is transient with environmental flux. Our observation that winter-adapted barley varieties exhibited higher global *RBCS* expression than their spring-adapted counterparts further suggests an environmental impact on RuBisCO evolution. Lastly, we identified 59C, located in the βA-βB loop region of RBCS, as a specific residue of interest for the future targeted modification of RuBisCO function, due to the observed positive effect of the G59C substitution on barley RuBisCO stability. Natural selection analysis led to the identification of residues under positive selection pressure in the βA-βB loop (of which a subset was in close proximity to the G59C substitution region). The results of this study help develop a clearer picture of RuBisCO evolution in the grasses, revealing novel promoter and sequence regions as potential targets for enhancing photosynthesis. Such targets may significantly enhance catalysis in C3 rubisco complexes and warrant future functional validation, with the possibility for superior crops with improved biomass generation.

## Figures and Tables

**Figure 1 ijms-26-07424-f001:**
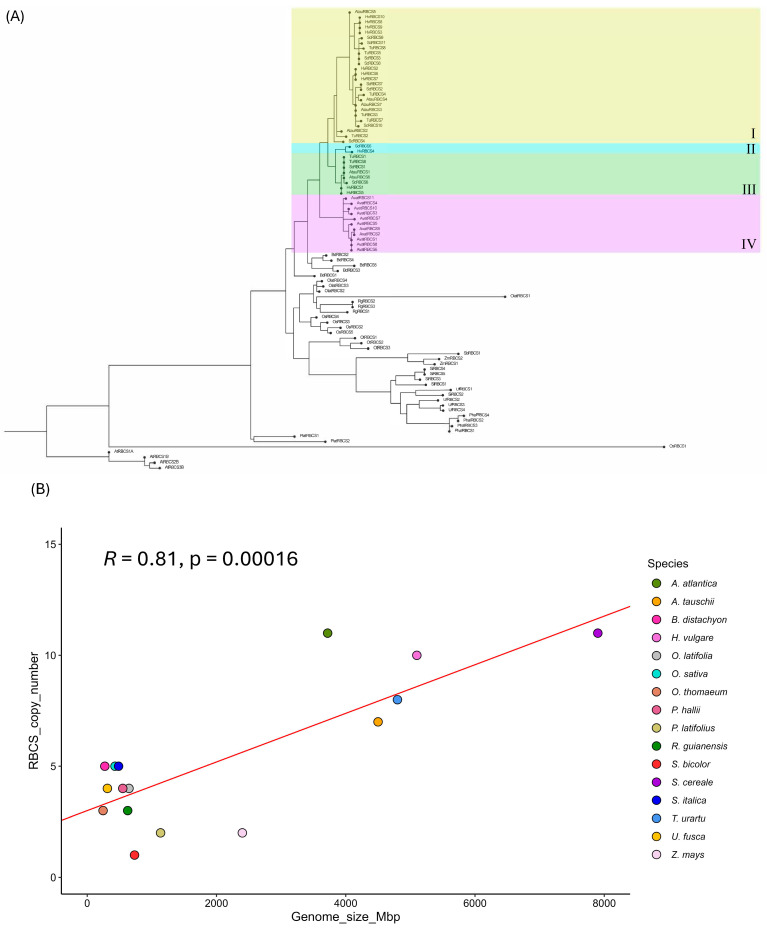
(**A**) Phylogenetic tree based on protein sequence similarity of *RBCS* sequences. Groups in *Triticeae* and *Aveneae* are highlighted. Group I: all *RBCS* on chr2 (including *RBCS* on chr7 in *S. cereale*), Group II: all *RBCS* on chr1 (*H. vulgare* and *S. cereale* only), Group III: all *RBCS* on chr5, Group IV: all oat *RBCS* (diverged separately from *Triticeae*). Gene code identifiers for the respective gene(s) are in Appendix A. (**B**) Linear regression plot of *RBCS* copy number (*Y*-axis) and genome size (*X*-axis) in 16 members of the true grasses. Included are the Pearson correlation coefficient (*R*) and its *p*-value and linear regression line (red).

**Figure 2 ijms-26-07424-f002:**
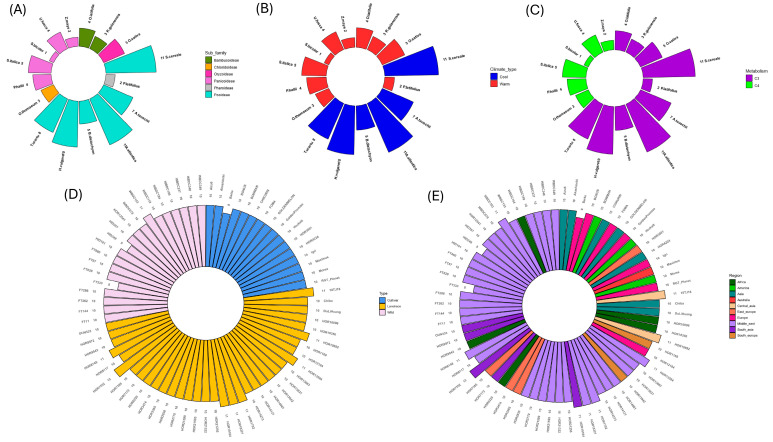
Polar bar plots show copy number variation in *RBCS* sequences between true grass species (*n* = 16) and are colourised based on sub-family (**A**), climate type (**B**), and metabolism (**C**). Pangenome copy number variation is plotted and colourised based on domestication status (**D**) and geographic region (**E**).

**Figure 3 ijms-26-07424-f003:**
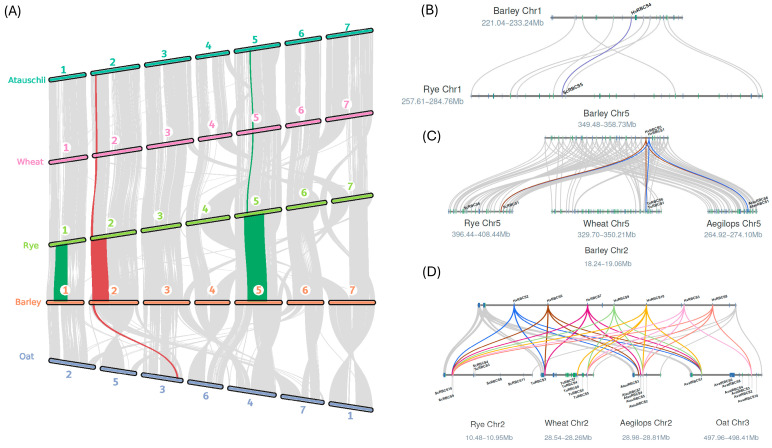
(**A**) Macrosynteny map showing syntenic blocks (colourised) containing RBCS genes shared between members of the *Triticeae* (*H. vulgare* (barley), *S. cereale* (rye), *A. tauschii* (Tausch’s goatgrass), *T. urartu* (diploid wheat)) and *Aveneae* (represented by *Avena atlantica*, a diploid oat). The red block indicates the region comprising syntenic *RBCS* gene copies shared across both members of the *Aveneae* and *Triticeae* tribes. The chromosome number of each species are labelled above. Microsynteny maps (right) display the syntenic relationship between barley and rye RBCS gene copies on chromosome 1 (highlighted in purple) (**B**), barley gene copies on selected regions of chromosome 5 (highlighted in blue and brown) with Rye, diploid wheat (*T. urartu*) and Tausch’s goatgrass/Aegilops (*A. tauschii*) (**C**), and finally between barley gene copies on selected regions of chromosome 2 (blue, brown, magenta, green, yellow, pink, and orange) with Rye (*S. cereale*), diploid wheat (*T. urartu*), diploid oat (*A. atlantica*) and *A. tauschii* (**D**). Syntenic *RBCS* copies are labelled above, noting that all maps use the barley chromosome as the reference for mapping. Numbers (presented as megabases, (Mb)) in the macrosynteny maps, denote the chromosome region being illustrated in the respective genomes.

**Figure 4 ijms-26-07424-f004:**
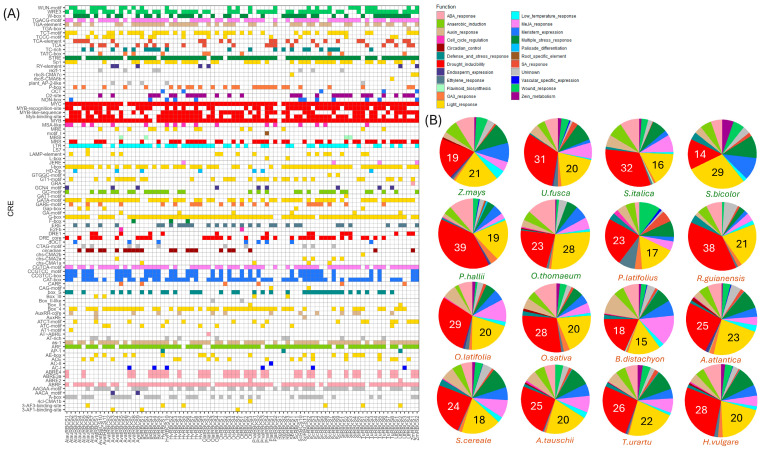
Presence-absence matrix displaying cis regulatory elements (CREs) present in *RBCS* gene family members derived from 16 true grass species, arranged by gene and colourised by CRE function (**A**). To the right (**B**) are pie charts demonstrating the relative percentages of CREs (based on function) that are present in the *RBCS* gene family of each species. A legend of CRE function colours is provided above.

**Figure 5 ijms-26-07424-f005:**
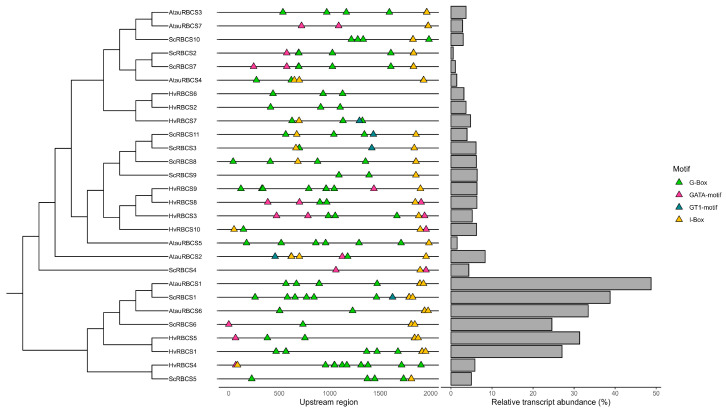
Panel figure demonstrating the relationship between specific promoter arrangements for *RBCS* gene family copies present in three members of the *Triticeae* (*H. vulgare*, *A. tauschii*, and *S. cereale*) (middle) and relative transcript abundance (given as a percentage) of *RBCS* gene copies (right). *RBCS* genes are grouped based on phylogenetic relationship, as shown by the cladogram on the left.

**Figure 6 ijms-26-07424-f006:**
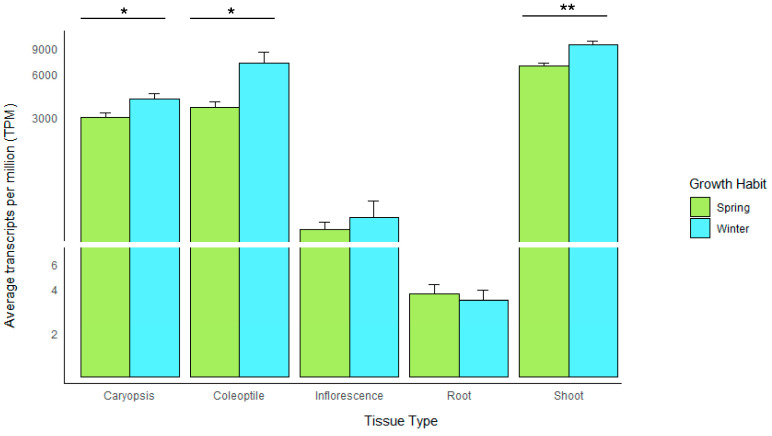
Barplot showing the average transcript abundance (TPM) for the *RBCS* gene family across five tissues for 19 barley pangenome accessions categorised by growth habit. Error bars are plotted and significance levels indicated for groups where there was a significant difference as determined by Student’s *t*-test. * = *p* < 0.05, ** = *p* < 0.01.

**Figure 7 ijms-26-07424-f007:**
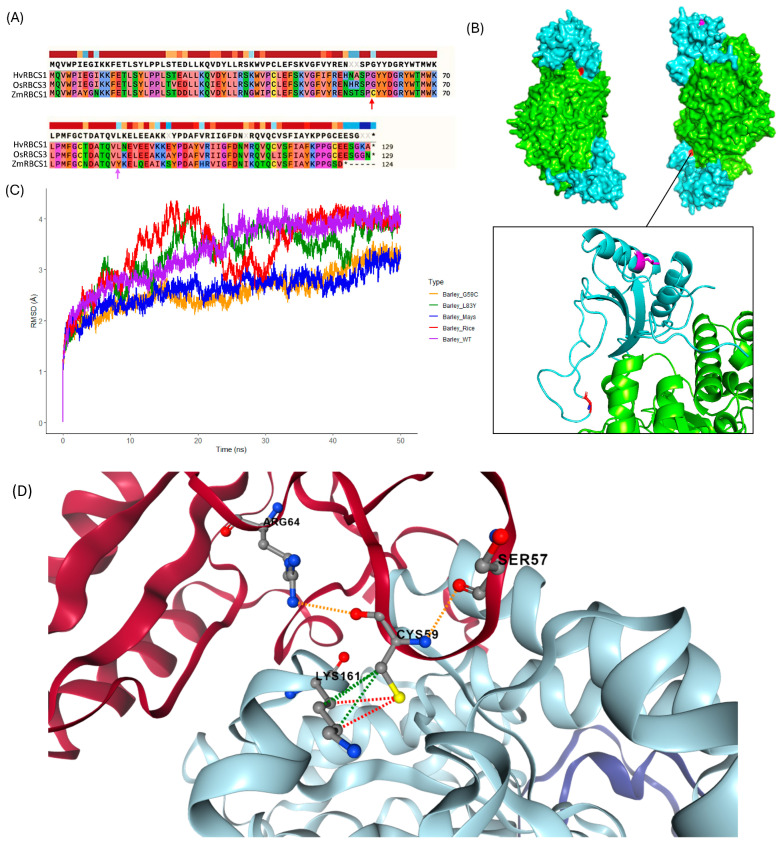
Protein alignment of Barley, Mays and Rice RBCS sequences, with residues of interest highlighted (arrows). Asterisks indicate termination of the protein sequence(s) (stop codon) (**A**). Residues at position 59 (red) and 83 (magenta) were selected due to combined close-proximity to the large subunit, distribution of amino acid types at these positions among the C3 and C4 grasses in this study, and the predicted impact of an amino acid substitution on stability of the barley RuBisCO protein (using structural and sequence-based predictors). Position numbering is based on RBCS derived from spinach (*S. oleracea*). (**B**) shows a protein surface model of the wildtype barley RuBisCO complex, consisting of two small subunits (cyan) and two large subunits (green), used in molecular dynamics simulations. Target residues covered in (**A**) are highlighted in red and magenta, representing G59C and L83Y substitutions, respectively (noting that these are the native residues at the selected positions in wildtype mays RBCS). A zoomed view of the G59 residue (Red) and the L83 residue (magenta) with their locations in the βA-Bβ loop (G59) and αA helice (L83) of the small subunit, is also provided in the plot above. (**C**) shows a line plot of relative Root Mean Square Deviation (RMSD) values (in Angstrom units (Å)) recorded over a 50-nanosecond timescale following molecular dynamics simulations of various hybrid RuBisCO complexes. Hybrid complexes were generated in silico using AlphaFold2 (ColabFold version 1.5.5) and PyMOL (software version 2.5.5) and include Barley RBCL complexed with Mays RBCS (Barley_Mays), Rice RBCS (Barley_Rice) and wildtype barley RBCS (Barley_WT). Two in silico mutants were also generated, with the production of Barley RuBisCO complexes possessing the single amino acid substitution L83Y (Barley_L83Y) and G59C (Barley_G59C) in the small subunit of RuBisCO, respectively. Molecular dynamics simulations were carried out using GROMACS version 2023.3. (**D**) shows a zoomed view of the G59C substitution (CYS59) displayed in DynaMut2. The red-coloured subunit is RBCS, whilst the light blue subunit represents RBCL. Interacting residues (with CYS59) are also labelled above. Dashed lines indicate polar (orange) and hydrophobic (green) interactions, whereas hydrogen bonds are represented in red.

**Table 1 ijms-26-07424-t001:** PAML (Phylogenetic Analysis by Maximum Likelihood) results of branch and branch-site-specific models, performed as part of the analysis of selection pressures in the RBCS gene family across select phylogenetic groups of true grass species. From left to right, the specified model, its number of parameters (np), likelihood value (lnL), the parameter value(s), and amino acid sites under positive selection following likelihood ratio tests (LRTs), are provided. Phylogenetic groups were defined as C4 group RBCSs (ω_C4_), Chr5 *Triticeae* RBCSs and relatives (ω_Ch5_), Chr2 *Triticeae* RBCSs and relatives (ω_Ch2_), and finally all other C3 group RBCS (ω_rest_). Two branch and site-specific models were implemented with C4 branch RBCS and Chr5 *Triticeae* RBCS defined as the foreground groups, respectively. The dn/ds ratios are represented as ‘ω’. Model A was defined to include four site classes: class I (ω_0_) and class II (ω_1_), representing conserved selection and neutral selection across RBCS lineages. Under the last two site classes, background lineages were defined to either possess ω_0_ or ω_1_ representing conserved or neutral selection, with the foreground lineage(s) possessing ω_2_, indicating positive selection pressure. The Null Model was defined by class I and II being the same as model A and the final site class differing, hence being defined by conserved or neutral selection across all lineages (ω_1_ = ω_2_). ‘P’ symbols indicate the relative proportions of site classes defined prior (e.g., p_0_ represents the proportion for ω_0_ and so on). Amino acid number positions are based on the barley reference sequence HORVU.MOREX.r3.5HG0468970. Analysis and table structure are based on the natural selection analysis method outlined in Jia et al. [25].

Model Type	np	lnL	Parameter Value(s)	Sites Under Selection
**One-ratio**				
*ω*_C4_ = *ω*_Ch5_ = *ω*_Ch2_ = *ω*_rest_	1	−8845.89	*ω*_C4_ = *ω*_Ch5_ = *ω*_Ch2_ = *ω*_rest_ = 0.04908	Not Applicable (NA)
**Branch-specific**				
*ω*_C4_ ≠ *ω*_Ch5_ = *ω*_Ch2_ = *ω*_rest_	2	−8841.77	*ω*_Ch5_ = *ω*_Ch2_ = *ω*_rest_ = 0.0479; *ω*_C4_ = 0.1999	NA
*ω*_Ch5_ ≠ *ω*_C4_ = *ω*_rest_ ≠ *ω*_Ch2_	3	−8843.84	*ω*_C4_ = *ω*_rest_ = 0.04908; *ω*_Ch2_ = 0.02328; *ω*_Ch5_ = 0.40993	NA
*ω*_C4_ ≠ *ω*_Ch5_ ≠ *ω*_Ch2_ ≠ *ω*_rest_	4	−8839.68	*ω*_rest_ = 0.04755; *ω*_C4_ = 0.20165; *ω*_Ch2_ = 0.02307; *ω*_Ch5_ = 0.44535	NA
**Branch and Site-specific (Foreground lineage: C4 branch RBCS)**				
Model A (assume positive selection on foreground)	4	−8802.28	*p*_0 _= 0.92048, *p*_1_ = 0.00571 (*p*_2_ + *p*_3_ = 0.0738); *ω*_0_ = 0.04573, (*ω*_1_ = 1.0), *ω*_2_ = 106.09735	34A (Probability > 0.7), 111R (Probability > 0.8), 7A, 8S, 35S, 36L, 37G, 70T, 102N, 103A, 143A (Probability > 0.9)
Model A Null (assume no positive selection on foreground)	3	−8810.99	*p*_0 _= 0.88145, *p*_1_ = 0.00543 (*p*_2_ + *p*_3_ = 0.11312), *ω*_0_ = 0.04575, (*ω*_1_ = 1.0, *ω*_2_ = 1.0)	NA
**Branch and Site-specific (Foreground lineage: Chr5 *Triticeae* RBCS)**				
Model A (assume positive selection on foreground)	4	−8818.62	*p*_0 _= 0.98593, *p*_1_ = 0.00602 (*p*_2_ + *p*_3_ = 0.00805); *ω*_0_ = 0.04726, (*ω*_1_ = 1.0), *ω*_2_ = 70.04518	103A (Probability > 0.98)
Model A Null (assume no positive selection on foreground)	3	−8820.80	*p*_0 _= 0.92467, *p*_1_ = 0.00560 (*p*_2_ + *p*_3_ = 0.06973), *ω*_0_ = 0.04723 (*ω*_1_ = 1.0, *ω*_2_ = 1.0)	NA

**Table 2 ijms-26-07424-t002:** Species selected for bioinformatic interrogation in this study. Included in the table are the metabolic type, subfamily and source of genome assembly for each species.

Species	Metabolism	Family/Subfamily	Assembly Source
*Pharus latifolius*	C3	*Pharoideae*	V1.1 https://phytozome-next.jgi.doe.gov/info/Platifolius_v1_1 (accessed on 22 March 2023)
*Oryza sativa*	C3	*Oryzoideae*	IRGSP-1.0 https://rapdb.dna.affrc.go.jp/download/irgsp1.html (accessed on 10 February 2023)
*Brachypodium distachyon*	C3	*Pooideae*	V3.1 https://phytozome-next.jgi.doe.gov/info/Bdistachyon_v3_1 (accessed on 25 February 2023)
*Avena atlantica*	C3	*Pooideae*	https://genomevolution.org; Maughan et al. [64] (accessed on 25 March 2023)
*Triticum urartu*	C3	*Pooideae*	Tu2.0 (IGDB) [64] https://plants.ensembl.org/Triticum_urartu/Info/Index (accessed on 25 March 2023)
*Secale cereale*	C3	*Pooideae*	Lo7_2018_HC https://wheat.pw.usda.gov/GG3/content/secale-cereale-lo7-files-2021 (accessed on 19 February 2023)
*Hordeum vulgare*	C3	*Pooideae*	MorexV3_2020_HC https://wheat.pw.usda.gov/GG3/content/morex-v3-files-2021 (accessed on 7 February 2023)
*Aegilops tauschii*	C3	*Pooideae*	Aet v5.0 https://www.ncbi.nlm.nih.gov/Taxonomy/Browser/wwwtax.cgi?id=200361 (accessed on 26 March 2023)
*Olyra latifolia*	C3	*Bambusoideae*	http://www.genobank.org/bamboo#2 (accessed on 4 April 2023); Guo et al. [65]
*Raddia guianensis*	C3	*Bambusoideae*	http://www.genobank.org/bamboo#2 (accessed on 4 April 2023); Guo et al. [65]
*Arabidopsis thaliana*	C3	*Brassicaceae*	TAIR10 https://phytozome-next.jgi.doe.gov/info/Athaliana_TAIR10 (accessed on 9 February 2023)
*Setaria italica*	C4	*Panicoideae*	V2.2 https://phytozome-next.jgi.doe.gov/info/Sitalica_v2_2 (accessed on 3 March 2023)
*Panicum hallii*	C4	*Panicoideae*	V3.1 https://phytozome-next.jgi.doe.gov/info/Phallii_v3_1 (accessed on 3 March 2023)
*Urochloa fusca*	C4	*Panicoideae*	V1.1 https://phytozome-next.jgi.doe.gov/info/Ufusca_v1_1 (accessed on 3 March 2023)
*Zea mays*	C4	*Panicoideae*	RefGen_V4 https://phytozome-next.jgi.doe.gov/info/Zmays_RefGen_V4 (accessed on 19 February 2023)
*Sorghum bicolor*	C4	*Panicoideae*	V3.1.1 https://phytozome-next.jgi.doe.gov/info/Sbicolor_v3_1_1 (accessed on 19 February 2023)
*Oropetium thomaeum*	C4	*Chloridoideae*	V1.0 https://phytozome-next.jgi.doe.gov/info/Othomaeum_v1_0 (accessed on 12 April 2023)

## Data Availability

Relevant data supporting this study is provided within the Appendix A and outlined in the article. Further inquiries regarding data can be directed to the authors.

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
