# Peer review of "A Comparative Bioinformatic Investigation of the Rubisco Small Subunit Gene Family in True Grasses Reveals Novel Targets for Enhanced Photosynthetic Efficiency"

_ijms, 2025, doi:10.3390/ijms26157424_

Round 1

Reviewer 1 Report

Comments and Suggestions for Authors

Rubisco efficiency is a central bottleneck in photosynthesis. Investigating the diversity and regulatory landscape of RBCS genes addresses a crucial research area with both fundamental and applied implications. This manuscript presents a comprehensive comparative analysis of the RBCS gene family across C3 and C4 grasses, integrating phylogenetics, synteny, cis-regulatory element profiling, expression analyses, and molecular dynamics simulations. The study identifies potential targets (e.g., G59C substitution) for engineering Rubisco with enhanced stability and efficiency, which is of significant importance for improving photosynthetic performance in temperate cereal crops. However, significant revisions are needed to improve clarity, methodological detail, and experimental validation.  

Language and clarity:
Many sentences are excessively long, particularly in the Abstract and Introduction (e.g., sentences extending over 5–7 lines), making them difficult to follow.

Experimental validation lacking:
Key conclusions, such as the proposed enhancement of Rubisco performance via G59C substitution and the effect of the terminal I-box pair, are based solely on computational predictions without experimental confirmation.
While molecular dynamics simulations are suggestive, in vitro or transient expression experiments would greatly strengthen the manuscript.

Interpretation of expression data:
In the analysis of barley pangenome accessions (Figure 5), significant differences in RBCS expression between winter- and spring-type varieties are reported. However, the physiological implications of these differences (e.g., potential correlations with photosynthetic rate or biomass accumulation) are insufficiently discussed.

Methods section details:
Important details of key analyses (e.g., parameters for cis-regulatory element prediction tools, molecular dynamics simulation settings such as force fields and simulation length) are missing, affecting reproducibility.
The sample size imbalance in the pangenome dataset (14 spring-type vs. 5 winter-type) should be addressed in terms of potential statistical power limitations.

Author Response

REVIEWER 1 comments:

Language and clarity: Many sentences are excessively long, particularly in the Abstract and Introduction (e.g., sentences extending over 5–7 lines), making them difficult to follow.

We have accordingly reduced sentence length where required in the abstract and introduction, with additional care taken to improve clarity with reduced sentence length in proceeding sections of the manuscript.

Experimental validation lacking: Key conclusions, such as the proposed enhancement of Rubisco performance via G59C substitution and the effect of the terminal I-box pair, are based solely on computational predictions without experimental confirmation.
While molecular dynamics simulations are suggestive, in vitro or transient expression experiments would greatly strengthen the manuscript.

We acknowledge that the above conclusions would certainly benefit from subsequent validation protocols – for example, future comparison of rubisco stability (as a result of the G59C substitution) would benefit from protein stability experiments involving such techniques as differential scanning calorimetry (DSC), which measures protein integrity following thermal exposure. In addition, measurements of enzyme efficacy (such as Kcat) would be beneficial to attain (as performed in a study by Matsumura et al. on a rice/sorghum rubisco hybrid) [Reference 20 in the manuscript reference list]. However, the validation process is not straightforward and would involve assembly of a mutant hexadecameric rubisco complex comprising wildtype HvRBCL and HvRBCS carrying the G59C substitution, and in the case of measurement of enzyme efficiency, would involve CRISPR-induced knockout of the large multigene RBCS family to observe the impact of the novel G59C substitution on photosynthesis. Similarly, investigative work on the impact of the terminal I-box pair would involve CRISPR-mediated removal of the identified CRE region and subsequent expression analysis. The combined analyses may hence take several months to over a year to sufficiently validate and are thus beyond the scope of our current manuscript. We believe the present bioinformatic analysis provides a potential avenue for photosynthetic improvement and would provide the wider academic community novel methods to explore this exciting pathway towards enhancing sunlight utilization in C3 crops. In addition, we have endeavored to highlight our identification of these novel photosynthetic targets as speculative at this stage. In turn we have now included protocol suggestions related to validation of the above I-box element pair arrangement and G59C amino acid substitution and in the section 3.2 and 3.3 respectively, of the discussion in this manuscript.

Interpretation of expression data: In the analysis of barley pangenome accessions (Figure 5), significant differences in RBCS expression between winter- and spring-type varieties are reported. However, the physiological implications of these differences (e.g., potential correlations with photosynthetic rate or biomass accumulation) are insufficiently discussed.

We thank reviewer 1 for their suggestion to strengthen the discussion, relating physiological impacts of the observed increase in RBCS expression for winter type barley varieties. In response to this comment, we have expanded section 3.1 in the discussion and have further explored the potential biological processes underpinning the relationship between enhanced winter type expression and photosynthetic rate/biomass generation.  

Methods section details: Important details of key analyses (e.g., parameters for cis-regulatory element prediction tools, molecular dynamics simulation settings such as force fields and simulation length) are missing, affecting reproducibility. The sample size imbalance in the pangenome dataset (14 spring-type vs. 5 winter-type) should be addressed in terms of potential statistical power limitations.

We have taken into consideration reviewer 1’s suggestions to improve clarity of the manuscripts methodology and have implemented the following changes. To improve parameters of cis-regulatory element (CRE) prediction tools, we have increased the detail outlining promoter map generation (i.e. CRE input file requirements, and organisation of input files in TB Tools) (section 4.4). The CRE data base PlantCARE does not include a parameter option, and the user only has the option of submitting a fasta file of query sequences to the database, so no changes were made to this section of the methodology. Molecular dynamics parameters (i.e. force field type and simulation length) were already present in the manuscript (methods section 4.7) – however, we have now included further detail on the choice of forcefield selection  (and model duration) and its suitability for modelling liquid systems containing complex protein heteromultimers (section 4.7). Increased detail has also been included in terms of parameter selection for molecular dynamics runs (section 4.7). In addition, we have now addressed the statistical implications of the small winter-type sample size in the discussion and its potential impacts on the observed results (section 3.1). Further expanding on the sample size discrepancy, unfortunately, there remains little available pangenome varieties to which transcriptome data is available (the pan-transcriptome data utilised in this study is the most up to date, only recently published in 2025: DOI:10.1038/s41588-024-02069-y). Thus, the lack of winter type pan-transcriptome representatives comes with associated sample size constraints.

Reviewer 2 Report

Comments and Suggestions for Authors

Dear Authors,

I have reviewed your manuscript. 

The manuscript presents a comprehensive bioinformatic and molecular dynamics analysis of the RBCS gene family in true grasses, with emphasis on Triticeae species. The topic is relevant and timely, particularly regarding strategies to enhance photosynthetic efficiency in crops. The introduction is well written, though slightly too detailed in the background sections.

The methods are appropriate and robust, but some bioinformatic steps (e.g. AlphaFold settings, selection models) would benefit from more clarity or referencing. The results are rich and well illustrated; the synteny analysis and expression divergence in barley accessions are particularly compelling. However, some figures (e.g. Figures 2 and 4) are overly complex and should be simplified.

The discussion is thoughtful, but at times overly speculative, especially regarding the proposed biotechnological applications of the G59C substitution and terminal I-box elements. A brief conclusion section summarizing the key findings would be helpful. Language is generally clear but would benefit from editorial tightening to improve readability.

Author Response

REVIEWER 2 comments:

The introduction is well written, though slightly too detailed in the background sections.

To improve clarity and flow of the introduction, we have accordingly reduced sentence length were required for enhanced readability. In addition, we removed a small subsection of the introduction covering ancient rubisco evolution, to improve overall conciseness in the presentation of background information pertinent to the study.

The methods are appropriate and robust, but some bioinformatic steps (e.g. AlphaFold settings, selection models) would benefit from more clarity or referencing.

We have now implemented key parameters used in the Alphafold2 modelling of the respective Rubisco complexes (section 4.7), including choice of model, pairing strategy, number of recycles, and maximum number of iterations. We thank reviewer 2 for this suggestion to improve methodological descriptions in the natural selection analysis section. Methodology for natural selection analysis has now been modified for improved clarity, particularly in the area describing likelihood ratio tests (LRTs) for confirmation of residues under positive selection (section 4.8). We note there was confusion between the Qi square test statistic and 2ΔL reported in the description of LRT calculations. These statistics were being used interchangeably in the methodology, when this is incorrect terminology. We have now fixed this discrepancy in the manuscript, such that only 2ΔL is referred to in calculations. In addition, further detail regarding selection parameters for the branch-specific and branch and site-specific model runs has now been provided in the methodology. Further detail is additionally provided in the supplementary material and includes all PAML code parameter files used in the natural selection analysis (file S4). In addition, we have cited a study by Jia et al. (2023) (reference 25 in the reference list) that utilised the same methodology, and to which our manuscripts methodology was adapted, which offers further clarification on experimental method if required by the reader.

The results are rich and well illustrated; the synteny analysis and expression divergence in barley accessions are particularly compelling. However, some figures (e.g. Figures 2 and 4) are overly complex and should be simplified.

Reviewer 2 has suggested reducing complexity of synteny maps (Figure 2 – now figure 3). We have implemented the following changes to improve plot quality and ease of interpretation for readers. First, we have modified labelled genes on the microsynteny plots from ‘gene_code’ identifiers to ‘gene_name’ identifiers as listed in supplementary file S1_gene_list_grasses (i.e. AA020104 changed to AvatRBCS1, and so on) and in doing so these gene labels are now consistent with other plots in the article. Gene labels colours have been changed to black for ease of readability. Syntenic links relevant to the study (coloured) have been brought in front of non-relevant syntenic links (grey) for increased contrast. This assists in reducing noise for microsynteny maps with overly extensive synteny relationships between non-relevant genes (in particular, Figure 3, C (chromosome 5 synteny maps), and better enhances the illustrated syntenic links between the labelled RBCS gene family members.

Given that the primary trend noticed in promoter element profiles of the RBCS gene family sequences, was the terminal I-box element pair within members of the Triticeae, we have removed panel (A) from figure 4 (now figure 5) and placed in the appendices. Panel A was removed given that both the promoter maps for all grass species and promoter element map for the Triticeae both illustrated the main findings, this being presence of the terminal I-box element pair in the Triticeae. The retained promoter map illustrates the observed terminal I-box element pair appropriately, while retaining conciseness. We have now included an extra supplementary file, File S5, containing the promoter map illustrating promoter element composition for all grass species in this study, that was originally included in the main manuscript (Figure S5.1).

The discussion is thoughtful, but at times overly speculative, especially regarding the proposed biotechnological applications of the G59C substitution and terminal I-box elements.

In response, we have modified the discussion in a manner that reduces the speculative nature of conclusions regarding the above findings. We have now incorporated a more conservative view of the G59C substitution in section 3.3, citing lack of experimental validation as a limiting factor, and have suggested future validation protocols. Similarly, we have noted the limitations of the perceived impact of the I-box element pair on RBCS efficiency, due to the computational nature of the findings, in turn suggesting avenues for functional validation of the cis regulatory region (section 3.2). We have additionally discussed potential limitations relating to the interpretation of pangenome expression data for spring and winter type barley varieties, further exploring possible explanations for observed RBCS expression differences, and highlighting the need for future experimental investigation (section 3.1).

A brief conclusion section summarizing the key findings would be helpful.

For improved conciseness, the conclusion section was modified to better summarise key findings – we specifically noticed that synteny analysis and natural selection analysis were inadequately summarised and have made changes accordingly.

Language is generally clear but would benefit from editorial tightening to improve readability.

We acknowledge that the manuscript presented lengthy sentence structure in areas and have endeavoured to shorten sentences where applicable for improved flow and readability.

Reviewer 3 Report

Comments and Suggestions for Authors

This manuscript provides a detailed bioinformatic and evolutionary analysis of the Rubisco small subunit (RBCS) gene family in grasses, combining pangenomic data, promoter architecture, expression profiles, and molecular dynamics simulations. The integration of structural and regulatory perspectives is a strength of the study and offers a multifaceted view on RBCS evolution in C3 and C4 species.

The introduction is extensive and informative, although it might benefit from some condensation to improve focus. Methods are generally sound, though a few analytical steps would benefit from more precise description (e.g. alignment thresholds, parameter settings for structural modeling). The molecular dynamics section is particularly interesting, though it would be helpful to include clearer justification for the selection of the specific amino acid substitutions tested.

The figures are informative but could be simplified in places; some are too densely packed with information, which affects readability. The proposed functional significance of the terminal I-box motif arrangement and the G59C substitution is intriguing but would need further experimental validation before drawing strong conclusions.

In summary, the study presents novel insights and adds value to the field of plant molecular physiology. With some clarification and modest editorial revisions, the manuscript will be suitable for publication.

Author Response

REVIEWER 3 comments:

The introduction is extensive and informative, although it might benefit from some condensation to improve focus.

In response we have modified sentence structure in the introduction section for improved clarity, as some sentences were overly long and impacted overall comprehensibility. We have also removed a subsection of the introduction that covered ancient rubisco evolution for improved flow of key background information.

Methods are generally sound, though a few analytical steps would benefit from more precise description (e.g. alignment thresholds, parameter settings for structural modeling).

Modifications have been made to the description of the protein structural modelling methodology, including further expansion on choice of forcefield selection, and simulation length. In addition, we have included additional details on the parameters selected for the molecular dynamics analysis (including selection of temperature and pressure control measures, and hydrogen bond constraints). The changes can be found at Section 4.7. Structural modelling parameters used for generation of proteins in Alphafold2 have also been provided in section 4.7. Alignment parameters used for the generation of ClustalO output, have now been reported in section 4.1 of the methodology.

The molecular dynamics section is particularly interesting, though it would be helpful to include clearer justification for the selection of the specific amino acid substitutions tested.

The results (section 2.6) covers justification of L83Y and G59C substitutions in detail, based on the results of protein alignment, and sequence/structure based prediction. However, we have now included additional justifications for residue selection that were initially omitted (for example, the presence of the aromatic side chain in the Y residue and its potential to impact protein stability), in section 4.7 of the methodology.

The figures are informative but could be simplified in places; some are too densely packed with information, which affects readability.

In response to reviewer 3’s comment, we have modified figure 2 (now figure 3) to enhance clarity of synteny maps, through modification of gene label length and colour improved readability. In addition, non-relevant syntenic links (grey) that were overlapping with syntenic links pertinent to RBCS gene family members, have now been placed behind the RBCS gene links (coloured) for ease of interpretation. In addition, we have improved the clarity of Figure 4 (now figure 5), through removal of panel A and placement of this figure in the supplementary material (supplementary file S5 – Figure S5.1). Both panel A and the condensed promoter map in figure 5 demonstrate the same major trend in promoter element composition that was found in the study (i.e., presence of the terminal I-box element pair in the Triticeae), and thus retainment of the condensed map improved conciseness in the presentation of results. Finally, we note that presentation of Figure 1 in its original state led to loss of detail in the phylogenetic tree and correlation plot. In turn we have split the phylogenetic tree and correlation plot into a separate figure to the polar bar plots, in turn allowing for enlargement of the tree and correlation plot figures for improved readability.

For other detailed figures (in particular figure 6 (now figure 7) and figure 3 (now figure 4)) we are concerned that further reduction in figure complexity will lead to loss of clarity in the results. For example, figure 7 shows the result of molecular dynamics simulations and provides insight into the location of the substitution, and its structurally relevant location in the RBCS subunit (i.e. within the catalytically relevant loop region). Figure 4 is difficult to condense given the analysis involved cross comparison of over 70 RBCS promoter sequences, and if simplified, resolution of promoter complexity and CRE element distribution would be lost.

The proposed functional significance of the terminal I-box motif arrangement and the G59C substitution is intriguing but would need further experimental validation before drawing strong conclusions.

We acknowledge that given the computational basis of our findings regarding the I-box motif arrangement and G59C substitution in our study, that we cannot yet formulate strong conclusions. As such, we have added emphasis on the computational limitations of our work in the discussion sections 3.2 and 3.3 regarding these two findings, and have in turn suggested future avenues for experimental validation.

Round 2

Reviewer 1 Report

Comments and Suggestions for Authors

The author's modifications are satisfactory, and the manuscript has reached publication standards.